# 'Ka asi kasya asi, kalyāṇi?' The Ambiguity of the *yakṣas* in the Araṇya Parva of the Mahābhārata

Arjan Sterken

Department of Comparative Religion, Radboud University, 6525 TH Nijmegen, The Netherlands;
arjan.sterken@ru.nl

**Abstract:** Supernatural entities are often described as ambiguous, but ambiguity is underdetermined and undefined. This article has a twofold goal: first of all, it constructs an ideal-type model for identifying and specifying ambiguity in supernatural beings; secondly, it examines the ambiguity of *yakṣas* in the Araṇya Parva of the Mahābhārata. This model for determining supernatural ambiguity utilizes five markers, which appear in either a positive or negative aspect: fulfilling or denying needs and desires; protecting or attacking humans; belonging to the same order as humans or rejecting this order; beautiful or hideous appearance; and living close by or far away from human communities. Four narratives are examined: the story of Nala and Damayantī, the First and Second War of the *Yakṣas*, and the story of the Drillling Woods. In all stories, each of the five markers are utilized to describe the *yakṣas*' ambiguity. However, one should distinguish between ambiguity proper (when conflicting markers are present at the same time) verus ambiguity caused by the shifting of markers during a narrative.

**Keywords:** Hinduism; Mahābhārata; mythology; folklore; *yakṣas*; ambiguity; Monster Theory

## 1. Introduction

The supernatural is the realm of the marvelous; the extraordinary. It can fill us with awe and a sense of wonder. *Yakṣas*, a species of supernatural beings found predominantly on the Indian subcontinent are sometimes regarded in that same light. As an example, in Mahābhārata 3.61:113–116 Damayantī, the human wife of King Nala, wanders alone in the forest, lost and forlorn. A caravan picks her up, and people start asking her questions:

> 'Who are you, whose are you, good woman? What are you seeking in the woods? The sight of you disturbs us, for are you human? Tell the truth, are you the *devatā* of this forest, or mountain, or region, good woman? We seek mercy from you! Are you a *yakṣī*, a *rākṣasī*, a noble woman? In any case, bring us luck, blameless woman, and protect us. Ordain, good woman, that this caravan safely depart from here, we seek your mercy!'[1]

While her sight disturbs the people from the caravan, Damayantī is also identified as a goddess (*devatā*), a noble woman (*vāraṅganā*), a *rākṣasī* (another type of supernatural beings), and a *yakṣī* or *yakṣiṇī*, a female *yakṣa*. So, even though she is found to be disturbing, she is still positively evaluated as a good and therefore luck-providing (*kalyāṇi*) and blameless woman (*anindite*); or perhaps it is wished that she is such a woman. This falls into a trend in which beautiful people, and especially women, are considered to be *yakṣas* or *yakṣiṇīs* (Misra 1981, pp. 31, 149); this is also found in narratives about Yayāti, Nala, Kirāta, Hanumān, Gangā, and Sītā (Misra 1981, p. 28). When identified with these benevolent human characters, those supernatural beings are also not that scary.

At other times, however, the supernatural is dreadful and terrifying. A little bit later in this narrative (a later interpolation in 3.62), after the caravan has gone through much misfortune, some people start to blame Damayantī and her potential supernatural nature as a *yakṣiṇī*:

'That insane woman who joined this mighty caravan in a misshapen and scarcely human appearance, she is the one who caused this dreadful illusion. Most certainly, she is a terrible *rākṣasī* or a *yakṣī* or a *piśācī*. All this evil is her work, why would we doubt it? If we see that wicked destroyer of merchants again, that causer of immense suffering, we shall certainly slay her who harms us, with stones, and dust, and grass, and wood, and cuffs.'[2]

Here, Damayantī is terrible or causes fear (*bhayaṃkarī*); she is a maniac-like woman, insane or intoxicated (*nārī hi-unmatta*). She is misshapen or distorted in form (*vikṛtākārā*), as if she is scarcely human (*rūpama-amānuṣam*). Indeed, the merchants of the caravan begin to question whether she is either a *rākṣasī*, a *yakṣī*, or a *piśācī* (a flesh-eating ghoul). This time, the *yakṣī* (and Damayantī) is not wonderful, but is instead terrifying.

I do not intend to suggest that Damayantī is a *yakṣiṇī*. It is quite clear by the end of the narrative that she is an exceptional human specimen. What is relevant here, is that she is potentially identified as a *yakṣiṇī* (or *devatā*, *rākṣasī*, or *piśācī*) by characters in the narrative. It tells us something about Damayantī's appearance in those specific instances, but also about the *yakṣas*, who can likewise appear as potentially blissful or potentially harmful. It seems, therefore, that there is no clear-cut image of these *yakṣas*. They are wonderful and dreadful at different times, and in that sense exemplify the *mysterium tremendum et fascinans* that Rudolf Otto attributed to supernatural beings.[3]

These *yakṣas* are, in a word, ambiguous. Ambiguity seems to be one of the key characteristics of the supernatural.[4] Supernatural entities are powerful, and possess skills not found among the human population (like providing good fortune or fertility to land, cattle, and people). It is often unclear, however, how willing they are to help mere humans. Sometimes they can be more inclined to tease humans, or downright scare or exploit them. This makes the supernatural tricky to deal with: you never know what any given encounter will yield. Gods are often conceived as portraying the best of humanity, while monsters represent the worst of us; but oftentimes beings that are considered gods could easily be monsters and vice versa (Laycock and Mikles 2021, pp. 3–4, 7). As Laycock and Mikles write, "sometimes all that separates a god from a monster is a dedicated PR team" (p. 4).

While Hiltebeitel (2003, p. 117) and Katz (1989, pp. 112–13) have noted that especially the human actors in the Mahābhārata are morally ambiguous, the same can be said of the supernatural actors, as the citations above indicate. Similar observations about the ambiguity of the *yakṣas* have been noted by Sutherland (1991, pp. 1, 51–52), Coomaraswamy (1971b, p. 1), Gonda (1960, pp. 323–24), and Misra (1981, p. 160). In this article, this samen ambiguity is examined with regard to narratives featuring the *yakṣas* in the third book (Vana or Araṇya Parva) of the Mahābhārata.

## 2. Theoretical Frame, Definitions, and Methods

### 2.1. Ambiguity

As stated above, supernatural entities are ambiguous. Ambiguity is a state of indeterminacy and ambivalence. That which is ambiguous cannot be precisely defined. Giesen refers to it as inbetweenness and fuzziness which defies categorization, meaning that it threatens social order (Giesen 2018, pp. 788–89; see also Kristeva 1982, p. 4). At the same time, this ambiguity is also constitutive of the social order, since strict categorization often does not fit reality (Giesen 2018, p. 792). Ambiguity is mainly the terrain of monsters in Monster Theory (also known as teratology) (As noted by Campbell 1996, p. 218; Cohen 1996, p. 6; Compagna and Steinhart 2019, p. ix; MacCormack 2013, p. 293; and Uebel 1996, p. 266), since they are beings which enable us to reflect on norms an anomalies by means of their appearance and given meanings (As noted by Cohen 1996, pp. 12–13; Friedman 2013, pp. xxviii, xxxvi; Mittman and Hensel 2018, p. x; Myhre 2013, p. 197; and Torrano 2019, pp. 132, 134). This can be extended to the monsters of religions as well, which are often dubbed as supernatural beings. For the purposes of this article the supernatural, while a tricky and Western-centric category, will be used for non- or formerly human beings with human-like intelligence, and often greater-than-human powers. Rather than

merely providing a meaningful and ordered cosmos, religious narratives actively generate ambiguity (Feldt 2012, pp. 1–3, 63), and supernatural entities play major roles in those religious universes by inhabiting ambiguous spaces, thus marking these spaces as special. These spaces are not safe by default; they could either harbour great rewards or great evil.

Ambiguity, next to indeterminacy, also denotes ambivalence. Ambiguous beings such as supernatural entities are not predetermined in their allegiances. They might help humans, or might harm them. Especially the fantastic elements in folklore and mythology play with these ambivalent and ambiguous tendencies by merging different dichotomies, like that between benign and malign, natural and supernatural, and self and other (Feldt 2012, p. 6). Such hybridity is seen as rather dangerous (Uebel 1996, p. 276); for it might uncover uncertainties about our conception of what is human (Friedman 1981, p. 3). Additionally, it seems clear that monsters, both literary and anthropological, are understood to reflect power relations, crises, inequalities, anxieties, and traumas (Musharbash 2014, p. 2). While this makes it seem as though monsters and the supernatural are predominantly malicious and troublesome, they actually seem to point to flaws within the cultures in which they appear, enabling us to resolve these issues (Cohen 1996, p. 20).

Ambiguity has never been properly conceptualized. Many articles and books assume ambiguity in supernatural beings, and demonstrate this by showcasing some tendency within the specific supernatural being in question. In order to demonstrate ambiguity in supernatural beings, I will propose a conceptual frame of when supernatural beings are positively evaluated and when they are negatively evaluated. I use the term 'evaluation' here to denote how the nature and behaviour of supernatural species are seen by humans. With this I do not intend to make a case for the existence of supernatural entities. At the same time, if one takes the highly valued methodological agnosticism of the scholarship of religion seriously, then I cannot make any statements on the matter of the ontological status of supernatural entities. I can only examine their phenomenological reality: people claim to have experienced their presence or influence, or at least tell stories about them (Laycock and Mikles 2021, pp. 10–12). That is why the human evaluation of their behaviour and presence is relevant.

My proposition is that when markers of these evaluations mingle within one supernatural species, then we are dealing with ambiguity. In this analysis, I will be relying on Max Weber's conceptual technique of the ideal type. The ideal type is an idealizing abstraction from reality based on many diffuse but concrete individual cases. Such an ideal type is not something found in actuality, but provides us with clear concepts which can be used to examine reality (Grønning 2017, p. 1; Weber 1904, pp. 64–65). In an ideal type, certain features of a phenomenon are made more visible and intelligible in order to demonstrate the unique qualities of the phenomenon in relation to other phenomena (Cahnman 1965, pp. 269, 271; Swedberg 2018, p. 184), while simultaneously providing it with a generic structure useful in comparative work (Cahnman 1965, p. 271). In doing this, an ideal type can demarcate separate features of a phenomenon which in otherwise are mixed up and indistinguishable (Hill 1973, pp. 150–61). Ideal types should not be seen as averages of reality or models of how reality should be, but can only be used in comparison with reality (Swedberg 2018, p. 184; Weber 1904, pp. 72, 76; Weber 1922, p. 10). The ideal types are also not hypotheses, but can be used in constructing hypotheses; this being the case, they serve as tools with which to do research, and are not the result of research (Segady 2014, p. 358; Weber 1904, pp. 64, 67). In comparison with reality, the ideal type helps to establish divergences or similarities, describe them with unambiguous concepts, and understand empirical reality rationally (Weber 1949, p. 43).

In essence, one will never find a purely positively evaluated supernatural being in reality, nor a solely negatively evaluated one, as they are Weberian ideal types. These ideal types have been constructed by referring to many empirical case studies of supernatural beings and their evaluation (see below). Since these case studies reflect a wide span of different cultures, we could establish these two ideal types of the positively and negatively evaluated supernatural entities as a heuristic means for exploring supernatural ambiguity

worldwide. In order to do this, one needs to append the ideal type model with concepts from the culture pertaining to each case study. In doing this, I am constructing a more sociological kind of ideal type (based on many examples) than a historical one (based on one historical period or society) (Hekman 1983, pp. 124–25). One of the main criticisms on ideal types is that they do not conform to reality (see Eliaeson 2000; Hekman 1983), which is actually a feature of the technique as stressed by Weber (1922, p. 10) Because of their heuristic nature, moreover, ideal types are not in constant need of empirical verification (Cahnman 1965, pp. 270–71). The reductive nature of ideal types is a problem with all kinds of modelling, since reality cannot be reproduced in a scientific model (Eliaeson 2000, p. 255). Segady and Svedberg rightfully state that ideal types are a necessary tool for the social sciences, while they also warn against ever seeing the ideal type as an actuality, especially after its utilization in research (Segady 2014, p. 358; Swedberg 2018, p. 184).

We can determine the ambiguity of supernatural entities, however, by how they score in different markers. The ideal type model provided here is a heuristic tool for exploring concrete case studies dealing with supernatural entities. Regarding the small data set, it is impossible to evaluate the usefulness of this model. Other research has contributed to demonstrating the heuristic value of the model (Sterken forthcoming), and future research within my PhD project at the Radboud University Nijmegen will establish its applicability more thoroughly through three different case studies. For now, the model is merely introduced and utilized. Scouring through the literature, five markers of positively evaluated supernatural entities can be found throughout the literature:

- It aids humans in fulfilling needs or desires, or helps them develop a means to them (Constructed from Bhutia 2019, p. 203; Bowyer 1981, p. 186; Cohen 1996, p. 16; Drewal 2013, pp. 78–79; Feldt 2012, p. 58; Jones 1944, pp. 246, 250; Kelley-Romano 2006, p. 397; Kieckhefer 1998, p. 15; Klaassen 2013, pp. 147, 151; Klaassen 2019, p. 21; Looper 2013, p. 211; Page 2011, p. 133; Parish 2015, p. 159; Petersen 2009, pp. 2, 13; Rockwell 1981, p. 43; Rose 1995, p. 150; Roth 2006, p. 46; Sontheimer 1989, p. 308; Starkey 2017, pp. 33, 38–39, 47–49; Waskul 2016, p. 10; and White 2003, p. 64);
- It protects humans against enemies or harm if called upon (Constructed from Bhutia 2019, pp. 194–95; Black 2020, pp. 147–48; Bloss 1973, p. 50; Doniger-O'Flaherty 1976, pp. 84–85; Erndl 1989, p. 239; Kelley-Romano 2006, p. 391; Klaassen 2013, p. 148; Kurlander 2017, pp. xi, 7, 53, 200, 277–80; Presterudstuen 2014, p. 133; Rose 1995, p. 152; Singh 2021, p. 122; and Starkey 2017, pp. 38, 45–47);
- It submits itself to the same kind of order to which humans do, or resides over that order (Constructed from Bhutia 2019, p. 193; Biardeau 1989, p. 31; Bloss 1973, pp. 38, 43; Borsje 1996, pp. 67, 75; Davidson 1981, p. 172; Davies 2013, p. 68; Felton 2013, pp. 107–22; Hafstein 2000, pp. 93–94; Hiltebeitel 1989a, p. 356; Kearney 2003, p. 42; Kurlander 2017, p. 7; Looper 2013, pp. 207, 215; Page 2011, p. 129; Riley 2005, pp. 275–76; Rockwell 1981, p. 46; and Shulman 1989, pp. 58–59);
- The experience of encountering the supernatural being (its appearance, smell, the emotional response to it, etc.) is culturally seen as pleasant or acceptable (Constructed from Borsje 2002, p. 75; Classen et al. 1994, pp. 42, 45, 47, 52–53, 104, 117, 130, 146; McHugh 2012, p. 79; Morton 2014, p. 79; Myhre 2013, p. 230; Sayers 1996, pp. 251–52; and Strickland 2013, p. 380);
- It inhabits spaces close to human civilization (Constructed from Bhutia 2019, p. 200; Hafstein 2000, p. 89; Klimkeit 1975, pp. 269, 279; Laycock and Mikles 2021, pp. 12–13; and Nugteren 2005, p. 13).

Similarly, the inverse markers denote negatively evaluated supernatural entities:

- It prevents humans from fulfilling needs, desires, or tasks (Constructed from Bullard 1989, p. 157; and Lancaster 1991, p. 278);
- It attacks or harms humans (Constructed from Ballard 1981, pp. 39–40; Beal 2002, pp. 62–63; Bhutia 2019, pp. 193, 195; Black 2020, pp. 62, 65–66, 148–49, 152, 155; Brown 1991, p. 14; Bullard 1989, p. 160; Carroll 1990, pp. 22, 42–43; Doniger-O'Flaherty 1976, p. 98; Erndl 1989, p. 239; Felton 2013, p. 104; Giesen 2018, p. 794; Jones 1944, p. 246;

Kurlander 2017, pp. xi, 281–84; Ling 1962, pp. 16, 20–21, 45; Looper 2013, p. 215; Mitter et al. 2013, p. 335; Mittman 2013, p. 8; Morton 2014, p. 78; Musharbash 2014, pp. 3, 5; Page 2011, p. 134; Pollock 1986, p. 271; Presterudstuen 2014, p. 133; Shulman 1989, pp. 48, 58; Singh 2021, pp. 121–22; Starkey 2017, pp. 33, 38, 42–45; White 2003, pp. 64–65; and White 2021, pp. 32–33);

- It tries to undermine the order to which humans submit themselves, or is generally contrary this order (Constructed from Asma 2009, p. 125; Beal 2002, pp. 6, 30; Biardeau 1989, p. 31; Black 2020, p. 216; Borsje 1996, pp. 7, 189; Borsje 2009, pp. 56–57; Braham 2013, pp. 17, 22–23; Carroll 1990, p. 34; Chalier-Visuvalingam 1989, pp. 171, 193; Cohen 1996, pp. 12–13; Compagna and Steinhart 2019, p. ix; Davies 2013, pp. 54–55, 68–70; Drewal 2013, p. 97; Dyrendal and Petersen 2012, pp. 217–19; Felton 2013, pp. 103, 105, 114; Friedman 1981, pp. 1, 3; Friedman 2013, pp. xxviii, xxxvi; Funk 2014, p. 144; Girard 1986, p. 13; Mittman and Hensel 2018, p. xi; Hiltebeitel 1989a, pp. 356, 361; Kearney 2003, p. 42; Kieckhefer 1998, p. 100; Kurlander 2017, pp. 55, 57; Li 2013, pp. 180, 195; Ling 1962, p. 16; Looper 2013, p. 197; Myhre 2013, p. 22; Petersen 2009, pp. 2–3, 12; Pollock 1986, pp. 271–72, 280; Presterudstuen 2014, p. 132; Riley 2005, p. 275; Shulman 1989, pp. 39, 48; Stasch 2014, p. 199; Steel 2013, p. 264; Strickland 2013, pp. 366, 370, 376, 383, 386; Tatar 2017, p. xxii; Torrano 2019, p. 134; Uebel 1996, p. 266; Van Duzer 2013, p. 388; Weinstock 2013, p. 276; and White 2021, p. 2);
- The experience of encountering the supernatural being (its appearance, smell, the emotional response to it, etc.) is culturally seen as disturbing or disgusting (Constructed from Alimardanian 2014, p. 94; Borsje 2002, p. 75; Carroll 1990, pp. 44–45; Cassaniti and Luhrmann 2011, p. 48; Classen et al. 1994, pp. 37–38, 54, 104, 117–19, 130, 149, 164; Cohen 1996, p. 6; Doniger-O'Flaherty 1976, p. 65; Feldt 2012, pp. 56, 60; Felton 2013, p. 104; Friedman 1981, p. 1; Giesen 2018, p. 795; Gilmore 2003, p. 41; Kieckhefer 1998, pp. 159–60; Lenfant 1999, p. 207; Li 2013, pp. 180, 182; Ling 1962, pp. 16, 45; Looper 2013, pp. 197–215; McHugh 2012, pp. 76, 79; Mitter et al. 2013, pp. 333, 335; Morton 2014, p. 79; Mukherji 2018, p. 113; Musharbash 2014, pp. 3, 8; Myhre 2013, pp. 222, 229–230; Riley 2005, p. 287; Pollock 1986, pp. 268–269; Sayers 1996, pp. 251–52; Starkey 2017, p. 35; Stasch 2014, p. 199; Strickland 2013, pp. 370, 380–84, 386; Watanabe 2020, p. 209; and White 2021, p. 138);
- It lives at the edges of human civilization or in the wilderness (Constructed from Asma 2009, p. 27; Borsje 1996, pp. 164, 168; Bullard 1989, p. 156; Davies 2013, p. 50; Feldt 2012, p. 251; Felton 2013, pp. 105, 123; Friedman 1981, p. 1; Friedman 2013, pp. xxviii, xxxiii; Frog 2020, pp. 455, 464; Funk 2014, p. 143; Kearney 2003, p. 3; Ling 1962, pp. 16, 20–21, 45; Lenfant 1999, p. 207; Manning 2014, p. 162; Musharbash 2014, p. 4; Myhre 2013, p. 220; Nugteren 2005, pp. 13–14; Pollock 1986, p. 270; Steel 2013, pp. 258, 261–63; Strickland 2013, pp. 366, 370, 386; Tatar 2017, p. xxii; Thurman 2014, pp. 30–31; Van Duzer 2013, pp. 387, 390–434; Watanabe 2020, pp. 206, 208; White 2003, p. 65; and White 2021, p. 9).

As stated above, the definitions of the above markers would be dependent upon specific cultural ideas and norms based on the data being analysed. In order to do that for the material considered here (the *yakṣas* in the Araṇya Parva of the Mahābhārata), we will examine ambiguity in Hindu traditions below.

## 2.2. Ambiguity in Hindu Traditions

Determining what ambiguity is within the Hindu context is challenging, since there are many contradictory ideas about evil in India, even within some of the selfsame texts (Doniger-O'Flaherty 1976, p. 19). Sutherland has similarly noted how most deities in India are surrounded by ambiguity (Sutherland 1991, p. 103). Often, however, one finds an extremely simplified and clear-cut delineation between good entities like the *devas* (gods) and *asuras* (demons),[5] but this does not hold true in the myriad Indian mythological traditions. *Devas* and *asuras* are not delineated by tendencies to help or harm humans (Doniger-O'Flaherty 1976, p. 63; Held 1935, p. 169). *Devas* are not representatives of the

good, nor are *asuras* invoked as explanations for evil in India; they are far too ambiguous to cause that. Rather, *devas* cause misfortune more often than *asuras* do (Doniger-O'Flaherty 1976, pp. 58, 141). While the Ṛg Veda presents *asuras* and *devas* in opposition to one another, it is unclear on what this actually entails (p. 57). Their opposition is certainly not moral (p. 58), but they do battle over world hegemony (Held 1935, p. 170). Held sees the conflict or contrast between the *devas* and *asuras* as the contrast between two moieties of a tribe (p. 171). Both of them are physically indistinguishable, and can assume various forms at will (*kāmārupin*) through the power of *māyā* or illusion (Doniger-O'Flaherty 1976, p. 62). Next to that, both species are related to each other as half-siblings. Both share Prajāpati as their father, while having different mothers (Held 1935, p. 169).

There are some differences between *devas* and *asuras*. While *devas* are active during the day, *asuras* and other beings like *yakṣas* and *rākṣasas* are active at night (Doniger-O'Flaherty 1976, p. 60; Held 1935, p. 169). Another distinction is power, and when *asuras* become too powerful, they must be destroyed so the *devas* can keep their hegemony (Doniger-O'Flaherty 1976, p. 63). Only in later times did *asuras* become hideous and immoral (p. 65). *Asuras* also tend to take on false doctrines (from Brahmin perspectives), while *devas* stick to the frameworks of Brahmin orthodoxy (Doniger-O'Flaherty 1976, p. 74; Sutherland 1991, pp. 185–88, 286–87). Lastly, the *devas* always win in the end (Doniger-O'Flaherty 1976, p. 59; Katz 1989, p. 32; Van der Velde 2007, p. 165), and the war between the *devas* and *asuras* will always continue (Doniger-O'Flaherty 1976, p. 59).

Ambiguity is a factor determined by humans, and its application to supernatural entities is necessarily influenced by the relations that humans have to the supernatural entities in question. Throughout different constellations of Hinduism, the dynamics between humans, *devas*, and *asuras* have shifted. In Vedic sacrificial religion, humans were allied with the *devas* against the *asuras* (Doniger-O'Flaherty 1976, pp. 79, 86). Especially Brahmin priests side with the *devas*, since the *devas* always win (Doniger-O'Flaherty 1976, p. 64). In post-Vedic asceticism, however, humans were sided with the *asuras* and other 'demonic' beings like *yakṣas* (all inhabiting the *āraṇya* or wilderness) in conflict against the *devas*, since ascetics evoke the wrath of the *devas* owing to their acquired power (*tapas*) (Doniger-O'Flaherty 1976, pp. 79–82, 86). This shift has to do with the competition between Brahmins and ascetics, who both claimed privileged access to the *devas*. According to the Brahmins, the power of the ascetics needed to be diminished (Doniger-O'Flaherty 1976, pp. 74, 80–82). In the *bhakti*-constellation, however, good men and good *asuras* were protected by the *devas* (especially Śiva and Viṣṇu), and the evil men and *asuras* were naturally at war with the *devas* (Doniger-O'Flaherty 1976, p. 82). At this point, men and *devas* become united in striving for *mokṣa* (Doniger-O'Flaherty 1976, p. 83).

Ambiguity also arises because certain questionable acts of the Brahmin priests and *devas* are justified in certain texts, because they allow those priests and *devas* to maintain their hegemony. As an example, a bad priest of the *devas* is acceptable, since anything is allowed that will tip the balance in the battle against the *asuras*. In the post-Vedic Hindu constellation, a good priest can shift alliance in order to rob good *asuras* of their powers. In the bhaktic constellation, priests will bring *asuras* to the *devas* as devotees (Doniger-O'Flaherty 1976, p. 138). Brahmins and *devas* deal with *asuras* by waging war against them in the Vedas up to the Purāṇas, and from the Brāhmaṇas to the Purāṇas, by means of barring their access to sacrifices (pp. 174–75).

The relationship between *devas*, *asuras*, and Brahmins does not immediately translate to other human populations. Wendy Doniger-O'Flaherty, in studying the *Yogavāsiṣṭha*, notes the various ways in which *śūdras*, women, and demons are depicted as both valuable and dangerous; or, ambiguous. While these three are often rejected by Brahmin orthodoxy, they can be highly valued in ascetic Hinduism. In addition to this, women represented seduction and illusion (*māyā*), while simultaneously being able to instruct how best to eradicate illusion—and demonic women brave even more of this ambiguity. Demons eat human flesh, but also seek superhuman knowledge, and in that sense became analogous

with ascetics. While a positive evaluation is possible here, it is not always so: the association of demons and *śūdras* is always negative (Doniger-O'Flaherty 1984, pp. 160–65).

Considering these points, it is easy to see how ambiguity easily becomes a part of Hindu mythology, since we are dealing with several different Hindu traditions, each with their own values and points of interest. When we look at the five markers (in both its positive and negative instantiation), and also when taking the *yakṣas* into account, they appear in the following guises:

2.2.1. Aiding in Fulfilling or Denying Fulfilling Desires, Needs, or Tasks

Both the conventional positively evaluated supernatural beings (the *devas*) and the negatively evaluated (*asuras* but also beings like *yakṣas*, *nāgas* and the like) are able to aid in the fulfilment of wishes or needs. Such boons can be attained through offering sacrifices or acquiring *tapas*. Especially the *devas* are known for trying to circumvent the rewards of *tapas*, since it threatens their hegemony. Instead, they try to offer other boons to the practicing ascetic, or make them lose their ascetic focus by tempting them with supernatural beauty like *apsarases*.

*Yakṣas* are known for granting certain benefits. They can be useful in agricultural contexts by providing rainfall and thunderstorms, but also more generally in that they can conjure up food, create baths, provide good fortune, impart knowledge, award wealth, immortality, and offspring (Gonda 1960, pp. 323–24; Misra 1981, pp. 3, 101, 150–51, 156–59, 163; Sutherland 1991, p. 54; White 2021, p. 105). At the same time, *yakṣas* are also known for stealing jewels instead of just providing riches (Misra 1981, p. 29), and especially *yakṣiṇīs* are known for eating children instead of providing them (p. 157). Also, greed and lust are seen as bad, since they appear after the *kṛta yuga*, meaning after the first age of a *mahayuga* during which everything is perfect (Doniger-O'Flaherty 1976, p. 29). Eating to resolve hunger is not necessarily evil, but it is when one eats improperly (Doniger-O'Flaherty 1976, pp. 32, 58).

Gaining things from *yakṣas* can be achieved through sacrifice or though Tantric practices. In terms of sacrifice, the pacified (so acting positively *yakṣa* enjoys the fragrance of jasmine and lotus and other fragrant things, the appearance of garlands of red and white flowers, cooked cereals, fruit, water, fish, flour cakes, and honey, and the performance of dance, song, and music (Agrawala 1970, p. 185; Misra 1981, pp. 98, 100). At the same time, especially in their negatively evaluated form, they can also enjoy liquor, flesh, and blood, items which are often tabooed for consumption (Misra 1981, p. 35; Nugteren 2005, p. 173).

*Yakṣas* and especially *yakṣiṇīs* are also heavily sexually connoted. *Yakṣiṇīs* often tempt human men sexually, which will have disastrous results if consummated. At the same time, while *yakṣiṇīs* are skilled seducers of men, *yakṣas* are not successful in seducing women, instead upholding chaste women and punishing promiscuous ones (Misra 1981, pp. 149, 157–58; White 2003, p. 64). Through Tantric rituals, *yakṣiṇīs* can be manipulated into becoming wives (Misra 1981, p. 56). Female sexuality is generally negatively evaluated, however. Lust is seen as evil, since it starts appearing at the end of the *kṛta yuga* (Doniger-O'Flaherty 1976, p. 29), and especially women can be seen as treacherous (Doniger-O'Flaherty 1976, p. 27; 2009, p. 233). Many cultures warn about female sexuality in the form of feminine monsters (Drewal 2013, p. 97; Li 2013, p. 180; Miller 2013). Sutherland reads the *yakṣiṇī* as a projection of Indian men onto women, who fear them because of the menace in case they are sexually unrestrained. Especially the lone wandering woman in the *āraṇya* (like Damayantī) will be accused of being evil or demonic; for good women are with their husband when outdoors (Sutherland 1991, p. 138).

Additionally, Sutherland describes that obstacles which prevent access to the *devas* are evil. Such obstacles can happen temporarily during initiatory or liminal situations, which for Sutherland are demonstrated in the function of the *yakṣas* as gate guardians (*dvārapālas*) of temples (Sutherland 1991, pp. 158–59). While here one can perceive them as protectors of the right order, they are also known to disturb rituals, especially the *śrāddha* offerings to the *pitṛs* (Misra 1981, p. 32; Sutherland 1991, p. 165).

Lastly, rebirth as a *yakṣa* is sometimes glorified. The rebirth as *yakṣa* can be achieved by virtuous people and animals (Gonda 1960, p. 323; Misra 1981, p. 147), as well as by fallen soldiers (Misra 1981, p. 28). However, rebirth as a *yakṣa* can also be attained as a punishment for breaking one's vows, wishing spiteful things, through an untimely death, or through evil acts (pp. 147, 159). Rebirth of human women as *yakṣiṇīs* is often regarded in this light. Sutherland states that Indian folk belief holds that women have reproductive needs. If those needs are not fulfilled, then a woman turns into a demonic *yakṣiṇī*, *nāgīṇi* (female *nāga*), or *rākṣasī* (female *rākṣasa*). This also happens when the passions and jealousies of women interfere with their social duties. Spirit cults can be established, or certain rituals performed, in order to prevent or pacify the hauntings of these demonic women (Sutherland 1991, pp. 145–47). A similar theme is found in Indian movies. In movies, lower-caste women or minority-caste women often turn into *yakṣiṇīs*. They are blood-thirsty ghosts who are wronged by high-caste men before their death, and therefore hunt men and drink their blood. Next to that, they are also sexually attractive (Chitra 2020, pp. 52–53). At the same time, it is considered a curse when *yakṣas* become human, since they lose their immortality (Misra 1981, p. 54).

### 2.2.2. Protecting or Attacking Humans

Suffering experienced in life or death is considered evil in manifold Hindu traditions, as is abusing one's own power (Sutherland 1991, p. 158). *Yakṣas* can enhance this suffering. Hopkins denotes their ambiguity in their double function of guarding and injuring (Hopkins 1915, p. 38). Agrawala and Misra furthermore state that *yakṣas* are demonic in the *Upaniṣads*, *Sūtras*, and *Purāṇas*, while they are protectors in the Atharva Veda, Tantric sources, and Jainism (Agrawala 1970, pp. 167, 188; Misra 1981, pp. 19, 26, 32). It is also uncertain whether they will help or harm humans (Misra 1981, p. 152). They are known to abduct people, murder them, rape them, eat them, steal from them, kill their offspring (Misra 1981, pp. 3–4; Sutherland 1991, p. 54), and cause diseases (Coomaraswamy 1971a, p. 5; Misra 1981, pp. 75–76, 150–55). However, they are also known to cure diseases (Gonda 1960, pp. 323–24; Misra 1981, p. 163; Sutherland 1991, pp. 166–67), and are also known as guardians of places and people (Agrawala 1970, pp. 167, 188; Bloss 1973, p. 38; Gonda 1960, p. 323; Misra 1981, p. 156; Sutherland 1991, pp. 120–21). Depicted on temple gates, they can serve as guardian deities (door guardians or *dvārapālas*; Misra 1981, p. 42; Sutherland 1991, p. 121), like at Bhārhut (Sutherland 1991, p. 106). They are also known as guardians of sacred fields (*kṣetrapālas*), sacred pools (pp. 121–22), and cities (Misra 1981, p. 159; Sutherland 1991, p. 146). Next to serving as guardians, *yakṣas* can aid humans in battles, often ensuring the victory of their side (Misra 1981, pp. 159–60). With this, destruction is not always evil (Doniger-O'Flaherty 1976, p. 58), since adhering to good behaviour also often leads to the destruction of the *asuras* (p. 130).

### 2.2.3. Conforming to or Destroying Human Order

Structural opposition to *dharma* is seen as evil in many Hindu traditions, and following *dharma* as good (Chitra 2020, p. 55; Sutherland 1991, pp. 2, 158). *Dharma* is both normative and descriptive: it describes how the world is, and tells how it ought to be (Doniger-O'Flaherty 1976, pp. 46, 94). This structural opposition would often be against *dharmaśāstra*, or the organization of society as envisioned by Brahmins (Nugteren 2005, p. 19). Within such a system, especially Brahmanicide is seen as the most heinous crime (Chalier-Visuvalingam 1989, p. 157). The *yakṣas* represent opposition to *dharma* by being opponents of the Pāṇḍavas in the *Mahābhārata* (Sutherland 1991, p. 158). Normally, when *dharma* is supported, the order of the natural world is maintained (Katz 1989, p. 31), meaning that opposition to *dharma* disrupts the natural order. From this point of view, it is evil to oppose *svadharma* (bound by *varṇa* and *āśrama* (stage of life)) in favour of *sanātana* or eternal *dharma* (Doniger-O'Flaherty 1976, p. 95); but such evaluation is reversed in post-Vedic asceticism. For Katz, the conflict between *devas* and *asuras* in the *Mahābhārata* is a conflict between *dharma* and *adharma*, where the *devas* and Pāṇḍavas fight to maintain *dharma* against the Kauravas, who are

*asuras* incarnate (Katz 1989, pp. 32, 48n17, 112–13). *Asuras*, do not disturb their *svadharma* by being evil in this framework, since their *svadharma* entails opposing the *devas* and killing humans (Doniger-O'Flaherty 1976, pp. 98, 130). Sutherland states that in the Purāṇas, both the *yakṣas* and the *rākṣasas* are ambiguous devices used to explore the opposition between *sanātana dharma* (eternal or universal *dharma*) and *svadharma* (*dharma* belonging to an individual's caste) (Sutherland 1991, p. 55).

Next to that, it is also evil to prevent the correct performance of rituals (Shulman 1989, p. 48; Sutherland 1991, p. 158). In Vedic Hinduism, this mainly consisted in the prevention of sacrifices from reaching the gods, while in Purāṇic sources it is especially the impediment of worship and access to temples (Doniger-O'Flaherty 1976, p. 183). *Yakṣas* do this by polluting physical elements during a ritual; by claiming the rewards of the sacrifice for themselves; and by falsely receiving gifts and worship (Sutherland 1991, p. 158). From the Brahmin point of view, ascetic practices also disturb the ritual order, and in Purāṇic Hinduism the *devas* are afraid of their own decline owing to the ascetic rise of humans. In order to stop this, the *devas* seek to morally corrupt ascetics, often by stressing the tediousness of *dharma* (pp. 24, 82).

The disturbance of social hierarchies and relationships is also considered evil (Shulman 1989, p. 48; Sutherland 1991, pp. 136, 158). *Yakṣas* do this by the transgression of sexual (seducing humans) and dietary (eating human flesh) restrictions (Sutherland 1991, p. 159). Abandoning of *svadharma*, which also entails abandoning social hierarchies, is considered to be an immensely evil act by Brahmin orthodoxy (Doniger-O'Flaherty 1976, p. 81). At the same time, post-Vedic asceticism often broke with the ideal of *svadharma*, and stressed the absence of distinctions in order to promote the goal of *mokṣa*. Asceticism in this context erases the distinction between humans, *devas*, *asuras*, *yakṣas* and others, and their implicit hierarchy (Doniger-O'Flaherty 1976, pp. 82, 90; Nugteren 2005, pp. 20, 91–92). Post-Vedic solutions were to destroy the ascetic power (Doniger-O'Flaherty 1976, pp. 82, 90), or the godly attributes of both ascetics and 'demonic' beings (p. 137), while bhaktic solutions turned the human or 'demonic' being into a *deva* (Doniger-O'Flaherty 1976, pp. 82, 90; Sutherland 1991, p. 158).

As elsewhere, here the *yakṣas* are prone to be both positively and negatively evaluated. They are active during the night, the time during which the *devas* of the proper order are asleep (Misra 1981, p. 150). While they counter proper order in that way, they can also be used to reiterate the proper order by their participation in the juridical process. According to Misra, criminals could be sent to the trees in which *yakṣas* live. There they would either defend their innocence or when their verdict is pronounced. If lying during their defense or when proven guilty, they would be crushed between the *yakṣa*'s thighs (Misra 1981, p. 155). Here, the *yakṣa*, while being part of legal proceedings, is still seen primarily as the punitive aspect of the legal system, instead of its acquitting and regulatory aspect.

### 2.2.4. Appearance

Obscuring true appearances through *māyā* is considered a great evil (Sutherland 1991, pp. 2, 158). *Māyā* can make us do evil things without our knowledge or consent (Doniger-O'Flaherty 1976, p. 7). *Yakṣas* are well-known as shapeshifters who can also create illusions (Agrawala 1970, p. 170; Coomaraswamy 1971a, p. 7; Misra 1981, pp. 146–47, 150–51; Sutherland 1991, p. 138). Because of this, we find them described as both beautiful and fierce-looking (Sutherland 1991, p. 54), and therefore ambiguous according to my model. Especially *yakṣiṇīs* are praised for their beauty (Misra 1981, pp. 3, 54; Sutherland 1991, p. 54). Beautiful *yakṣas* are said to give off light (Misra 1981, pp. 1481–49) but are otherwise not described elaborately. Horrifying *yakṣas*, on the contrary, have many markers. They have red eyes which are squinted or do not blink, dark hairy bodies with coarse skin, pointy ears, a dwarfish stature with a hunched back, frightening faces, huge mouths, feet that are turned the wrong way, other features which resemble those of elephants, bears, and birds, and no shadow (Misra 1981, pp. 3, 32, 147–49, 158–59; Sutherland 1991, pp. 54, 59). In literary texts, there is no solid model for the appearance of *yakṣas* and *yakṣiṇīs*, and

one often finds diverging descriptions of them. They lose their illusory shape during sex, calamity, sleep, anger, fear, or ecstasy (Misra 1981, p. 147). In addition to shapeshifting, they can also turn invisible, which highlights their indeterminacy (Misra 1981, pp. 9, 147).

### 2.2.5. Location

Total Otherness or the unknown is often attributed to an enemy or threat (Sutherland 1991, p. 158). This can be Otherness in the term of human Others, but also in terms of geographical distance. To begin with the former, *yakṣas* together with creatures like *nāgas* and *rākṣasas* often resemble the tribal, the foreign, and the uninhabited—in short, everything that falls outside of the known order of villages and Brahmins (Doniger-O'Flaherty 2009, pp. 245–47; Sutherland 1991, p. 159). The image of the *yakṣa* can also be projected upon that of the Untouchable (or *dalit*), who often represent the savage and uncultivated in opposition to *kṣatriyas* (Doniger-O'Flaherty 1984, pp. 162–65; Sutherland 1991, p. 120). This does not mean, however, that *yakṣas* are always *dalit*: each individual supernatural being can belong to a different caste, and supernatural species are never uniformly placed within one caste (Sutherland 1991, p. 164).

Regarding the latter, *yakṣas* can also indicate the geographically distant. The Epics mark a tension between village life (*grama*) and the forest (*āraṇya*) (Thapar 2003, pp. 103–104). According to Nugteren, the Brahmin division of the local area is threefold. There is a division between the dharmic village (*grama*) where the Brahmins rule, and the adharmic forest (Nugteren 2005, pp. 11, 16, 85). The forest, however, is split up in two different areas. There are the woodlands surrounding the village which the village uses (*vana*), the so-called safe forest. Then there is the *āraṇya*, which is the forest that is beyond human and Brahmin control. This is the place where harmful beings reside, such as indigenous tribes, wild animals, outlaws, monsters, and *yakṣas*. At the same time, it is also the place where *soma* is found, where *tejas*- or *tapas*-filled *vrātyas* and *sādhus* wander who can bless and heal or curse and harm. While it is away from the village, the *āraṇya* is also an escape from *saṃsāra* (Nugteren 2005, pp. 12–14; Van Buitenen 1973, p. xxii). The *āraṇya*, next to being dangerous, also provides liberation from Brahmin order and their cycles of rebirth. The *āraṇya* is also the place of the *āśrama*, which especially in the Epic context is the place of exile or idyllic holiday (Nugteren 2005, p. 14). In addition to all this, especially in the Aranya Parva of the Mahābhārata, the *āraṇya* is the place where the Pāṇḍavas, through austere practices, encounter the *devas* who provide them with secret weapons (Berry 2022, pp. 75, 77). The *āraṇya*, in short, is ambiguous; a place of purity and monsters (Parkhill 1995, p. 8)

So *yakṣas* are often located within the *āraṇya*, often residing within a *caitya* or *āyatana* (Bloss 1973, p. 37; Misra 1981, p. 50), which is an open-air shrine typically found outside the city in a grove or on a mountain, or on the edges of settlements (Agrawala 1970, p. 189; Misra 1981, pp. 42, 89–90, 97). They can also live in forests, lakes, trees, deserted halls, or on mountains (Misra 1981, pp. 42, 89–90). Here they have most power, and are able to devour anyone who trespasses on their terrain (Misra 1981, pp. 150, 154). *Caityas* are not merely sinister spaces, but are also considered to be good resting spots for travelers and mendicants, especially in Jain and Buddhist sources (Coomaraswamy 1971a, p. 23), and are likewise signposts on the pilgrimage road itself (Sutherland 1991, p. 121), or even a pilgrimage destination (Misra 1981, p. 52). Sometimes *yakṣas* are found a bit closer to home, inhabiting the borders of towns and villages (Misra 1981, pp. 42, 89–90, 97; Sutherland 1991, p. 159). When worshipped, *yakṣas* will also guard the gates of shrines or cities (Coomaraswamy 1971b, p. 8; Misra 1981, pp. 89, 93). Next to that, *yakṣas* are also found as the tutelary deities of houses (Misra 1981, pp. 20, 93). Thus, White's statement that *devatā* inhabit grove *caityas* and *yakṣas* those in urban centres does not seem to be strongly substantiated (White 2021, p. 102).

### 2.3. Method

For this article, the Vana or Araṇya Parva of the Mahābhārata is analysed. The Mahābhārata is described as one of the two great Epics of India, together with the Rāmāyaṇa.

A tentative dating between 400 BCE and 400 CE is often accepted (Katz 1989, p. 2; Van Buitenen 1973, p. xxv). However, one has to take into account that the critical edition of the most ancient variants of the story contain substantial interpolations, meaning that there is no unifying artistic design behind them that was completed at a discrete point in time (Van Buitenen 1973, pp. xxiii-xxiv). Different instances of disunity have arisen throughout the years. Holtzmann identified different layers with different interests, like heroic *kṣatriya* epic, *brāhmaṇic* didactic passages, or devotional *bhakti* hymns (Holtzmann 1892, p. 8); this was already criticized while he was active (Hopkins 1892, pp. 500–501) while also partly reproduced in later scholarship (Katz 1989, p. 4). In addition to that, the text's main narrative is incorporated within multiple narrative frames (Shulman 2001, p. 29). While the Mahābhārata is often described as an epic, Hiltebeitel would argue for a multi-generic approach to the critical text (Hiltebeitel 2003, pp. 122, 132), containing multiple genres, voices, and even narratives. Katz and Shulman even consider whether the Mahābhārata is more appropriately considered an encyclopedia (Katz 1989, p. 9; Shulman 2001, pp. 26–28).

The Sanskrit edition which is utilized is known as the Pūna or BORI edition. According to Fitzgerald, it reflects the grand Mahābhārata synthesis of the 300–400 CE when the Gupta empire rose to power (Fitzgerald 2020, p. 4). McGrath states that the Mahābhārata became a source of political legitimacy starting with this dynasty, and is nowadays seen as the foundational myth of India itself (McGrath 2019, pp. 41, 83). The Mahābhārata itself is more than that text, however. Next to many regional variations, there are also all manners of plays, depictions, television shows (Hawley and Pillai 2021, pp. 29–30), and more found throughout the ages. As for translations, I use Van Buitenen's 1975 translation, and Ganguli's 1884 translation. The Sanskrit edition is used to check the translations and to suggest amendments (rare in Van Buitenen, more common in Ganguli). This serves to examine different attestations in different manuscript traditions, and to translate certain sections that have been neglected by Ganguli's otherwise complete translation of the Sanskrit Mahābhārata tradition. The Sanskrit passages will be provided in the footnotes, but only when the *yakṣas* actually appear in them. Other passages have been omitted. The different manuscript attestations have been referred to by the same system which is utilized in the BORI edition. A description of those manuscripts can be found in Sukthankar (1942, pp. i–x).

I will exclusively examine the *yakṣas* in the Araṇya Parva of the Mahābhārata, and how they are portrayed. 98 textual places have been located, which have been inductively coded through open coding with the program Atlas.TI. Some of these textual places will, however, not be analysed for this article. In order to maintain a proper length I will focus on four narratives within the Araṇya Parva of the Mahābhārata: the story of Nala and Damayantī (Section 3.1; 3.50–3.78); the First War of the *Yakṣas* (Section 3.2; 3.146–3.153); the Second War of the *Yakṣas* (Section 3.3; 3.157–159); and the story of the drilling woods (Section 3.4; 3.295–3.299).

## 3. The *Yakṣas* in Narratives in the Araṇya Parva of the Mahābhārata

In the Vana or Araṇya Parva of the Mahābhārata, the Pāṇḍavas are moved into the *āraṇya*, the wilderness, which is the terrain of the *yakṣas*. The Araṇya Parva is also that part of the Mahābhārata infested with the highest concentration of *yakṣas*. The Pāṇḍavas, having lost the pivotal game of dice against the Kauravas, are exiled into the *āraṇya*. The Parva includes a mixture of narrative action which prepares the Pāṇḍavas for the upcoming battle with their rivals by providing them with weapons and many homiletic and philosophical teachings, sometimes meant to prepare them for the upcoming war, but sometimes with quite different goals in mind (Bailey 2022, p. 42).

Previous analyses of the *yakṣas* in Epic sources have led to some overgeneralised statements. Misra states that *yakṣas* in Epic literature are mainly benevolent but sometimes uncanny, while they are malevolent in Pāli literature (Misra 1981, p. 28). Such a statement is not warranted for the myriad roles *yakṣas* play in the Mahābhārata, however. According

to Sutherland, the *yakṣas* are portrayed in Hindu mythology as opposition to kings or *kṣatriyas*. The *yakṣas* are savages and uncivilized, while kings represent the highest values and order of society (Sutherland 1991, p. 120). In Epic mythology, *kṣatriyas* are the ones who should establish control over displaced demons, rather than the *devas* (p. 53). In a sense, the encounters with *yakṣas* are used in the Epics and Purāṇas as a means to reestablish the sovereignty of the *kṣatriyas* in addition to their power (Sutherland 1991, pp. 121–22). While this statement might work for certain narratives (especially the First and Second War of the *Yakṣas* seen below), it is not a sufficient explanation for other passages. Below we will examine the *yakṣas* in more detail.

### 3.1. The Yakṣas in the Story of Nala and Damayantī

The story of Nala and Damayantī (3.50–3.78) is well-known to students of Sanskrit: the practice of translating this story as one's first real practice with Sanskrit was initiated by Caland. In the narrative, Nala, prince of the Niṣādas, and Damayantī, daughter of king Bhīma (not the Pāṇḍava), fall in love with each other by merely learning of each other's existence. Through a *svayaṃvara*, Damayantī is able to pick him as her husband. While happily living together, Nala ruins the kingdom by losing it in a game of dice, after which both of them enter into exile in the forest. At a certain point, Nala gets separated from Damayantī through demonic tricks, and Damayantī starts searching for him. After many adventures she manages to locate him, and Nala is able to win back the kingdom after another dice game, and becomes the king of the Niṣādas. This short and undetailed synopsis showcases the similarity to the Pāṇḍavas' situation at that point in their narrative, as they are freshly banished and need to live in the *āraṇya*.

This story has already been discussed at the beginning of this article. There are some extra textual places which warrant discussion in order to yield a complete picture. In total, there are six textual places in which the *yakṣas* appear in this narrative, of which we have already seen two. The other four are relatively brief and easier to characterize. In two cases, the *yakṣa* is a marker of beauty. First, in 3.50:13, Nala praises Damayantī's beauty, stating that no one among the *devas*, *yakṣas*, people,[6] and others have heard about or seen such beauty.[7] This verse insinuates that these beings (especially the *devas* and *yakṣas*) are normally beautiful, but that even they themselves have not seen a beauty like Damayantī. Likewise, in 3.52.16, Nala's beauty makes Damayantī question whether he is a *deva*, *yakṣa*, or *gandharva*,[8] insinuating that these beings are known for their beauty.

In the two other cases, the *yakṣa* Maṇibhadra is invoked. Maṇibhadra is one of the few *yakṣas* which receives a name in Indian texts. In the first textual place (3.61:123), Damayantī joins up with a merchant caravan during her search for Nala. Śuci, the leader of that caravan, calls upon the *yakṣa* Maṇibhadra to aid them in their search.[9] Maṇibhadra is indeed known by other sources to preside over caravan merchants (Agrawala 1970, p. 184). In the second textual place (lines 60 and 61 of a substitution by certain manuscripts of 3.62:1–17), the mood has shifted. Bad events have befallen the caravan, and its members believe that they are to blame: they failed to worship Maṇibhadra and Kubera (also called Vaiśravaṇa).[10] This latter figure is considered in many passages to be the king of the *yakṣas*.[11]

When we examine these six textual places which mention the *yakṣas*, we note some ambiguous usages of this figure. Let us therefore now examine it through the model of five markers as developed earlier. Concerning, the first marker describing the fulfilling of needs or preventing them, we see on the one hand that *yakṣas* can help humans (3.61:123) or bring luck (3.61:115). While this is a positive evaluation, we see a more ambiguous portrayal when examining the second marker, describing protection or attack. When first encountered, Damayantī, the potential *yakṣiṇī*, is asked to protect the merchant caravan (3.61:116). Later, however, she turns into a wicked destroyer of merchants and causer of immense suffering (line 63 of appendix 11 of the critical edition, a substitution for 3.62:6–10). In addition to this, Maṇibhadra and Kubera allow the caravan to be attacked because they have not been worshipped sufficiently (lines 60 and 61 of appendix 10 of the critical edition, a substitution of 3.62:1–17).

At the third marker which is concerned with the proper social order, we find that Damayantī is first approached as a proper woman: someone who is married and brings luck (*kalyāṇī*, 3.61:113; *varāṅganā*; and *anindite*, both verse 115). However, quickly it is determined that she is instead a figure which only brings *pāpa*, a moral and natural kind of evil (lines 62–63 ofappendix 11; Doniger-O'Flayerty 1976, p. 6). Additionally, Damayantī wanders alone, which is the mark of a wild woman, not tied to a husband, who could therefore be treacherous, monstrous, or both (Sutherland 1991, p. 138). Lastly, the caravan believes their misfortune is caused by breaking with order, by neglecting worship of Maṇibhadra and Kubera (lines 60–61 of appendix 10).

As for the fourth marker regarding appearance, we encounter ambiguity as well. The beautiful side of being supernatural or a *yakṣa/yakṣiṇī* is stressed three times (3.50:13; 3.52:16; and 3.61:115). However, this image is turned upside down in lines 58–60 of appendix 11, where the *yakṣī* is associated with other sinister supernatural beings like *rākṣasas* and *piśācas*, but where Damayantī, potentially identified as a *yakṣiṇī*, is called insane (*nārīva unmatta*), misshapen (*vikṛtākārā*), and scarcely appearing human (*rūpamamānuṣam*); her earlier, beautiful appearance was merely an illusion (*māyā*).

As for the final marker concerning location, one has to remember that the Araṇya Parva of the Mahābhārata mainly plays out in the *āraṇya* or wilderness, a place associated with danger in Brahmin orthodoxy. This is clearly seen in the ambivalence with which Damayantī is treated both in 3.61:113–116 and appendix 11. Even though in 3.61:113–116 the response to her is mainly positive or hopeful towards a positive resolution, the members of the caravan are disturbed by her sight (*vyathitāḥ*, verse 113), for she is a lone wandering woman in the *āraṇya*. First, they try to pacify her and ask her to help them and bring them fortune, but after misfortune has befallen the caravan, they turn their backs on Damayantī.

In short: while the location seems to be primarily marked as negative (since it is the far-off *āraṇya*) and the first encounter of Damayantī with the caravan is potentially positively evaluated by means of possible wish-fulfilment, all other markers show mixed characteristics, essentially demonstrating that the narrative of Nala and Damayantī utilize the figure of the *yakṣa* as an ambiguous figure (see Table 1).

**Table 1.** Evaluation of the *yakṣa* in the story of Nala and Damayantī.

| Marker | Positive | Negative |
|---|---|---|
| Help/Hindrance | Asked for help (3.61:123) or bring luck (3.61:115) | - |
| Protectors/Attackers | Asked to protect (3.61:116) | Destroyer and causes suffering (line 63 of appendix 11); allow the caravan to be attacked (lines 60–61 of appendix 10) |
| Social order | Good woman (*Kalyāṇī* (3.61:113), *varāṅganā*, *aninditā* (115)) | *Pāpa* (lines 62–63 of appendix 11); Damayantī wanders alone; no worship (lines 60–61 of appendix 10) |
| Beautiful/Gruesome | Beauty (3.50:13; 3.52:16; and 3.61:115) | Misshapen, illusion (lines 59–60 of appendix 11) |
| Central/Peripheral | - | *Āraṇya* |

### 3.2. The Yakṣas in the First War of the Yakṣas

The Mahābhārata, in the conglomerate shape available to us in the critical edition, abounds with repetitions. One of these duplicated stories is known as the *yakṣa-yuddha* or war of the *yakṣas*, found twice in 3.146–153 and 3.157–159. In short, both stories tell of how the Pāṇḍavas stay on a specific mountain during their exile. Draupadī, at a certain point, smells fragrant flowers (*saugandhika*), and sends out Bhīma to fetch her some. This will

bring Bhīma within the bounds of Kubera's territory, which is guarded by his *yakṣas* and *rākṣasas*. A battle ensues, and, while both sides fight valiantly, Bhīma eventually wins.

In total, the *yakṣas* are found in nine textual places in the first narrative, and in seventeen textual places in the second narrative. One finds less *yakṣas* in the first narrative because the name *yakṣa-yuddha* is a misnomer for this first conflict: the guardians of Kubera's domain are predominantly *rākṣasas* in this encounter. This is not surprising, since it has been previously noted that the *yakṣas* and *rākṣasas* are often mutually interchangeable (Misra 1981, p. 27; Sutherland 1991, p. 49). Similarly, Kubera is considered to be the king of *yakṣas*, *rākṣasas*, *gandharvas*, *guhyakas*, *nairṛtas*, and *piśācas* (Gonda 1960, p. 324; Misra 1981, pp. 5, 60). Van Buitenen believes the second story to be a correction of the first one by adding the *yakṣas* (Van Buitenen 1975, pp. 201–2).

Bhīma is the protagonist of both stories, and is generally known in the Mahābhārata as the slayer of *yakṣas* and *rākṣasas* (Misra 1981, p. 28; Sutherland 1991, p. 52). In the First War, Bhīma departs for Mount Gandhamādana. This mountain, whose name means 'intoxicated by perfume' (McHugh 2012, p. 94), is generally described as a lovely and beautiful place (3.146:20–33). Bhīma disturbs the peace of the area, killing many animals (3.146:38–48). Berry stresses that the beauty of the place signifies a kind of mythic environment, which can only be enjoyed upon paying the price of austerity or boldness (Berry 2022, p. 86). The beauty of the mountain is illustrated by some of its inhabitants like *yakṣas*, *gandharvas*, *devas*, and *brahma-ṛṣis* (3.146:23).[12] Some manuscripts replace these generally positively evaluated beings with more troublesome ones. B1 changes the *devas* (or *suras*) into *asuras*, while the Dc-manuscript group remove the *yakṣas* for the *rākṣas* or *rākṣasas*. The beauty of Mount Gandhamādana is further described in 3.146:32–33 where the wives of *yakṣas* and *gandharvas* stare at Bhīma.[13] Manuscript D2 replaces the *gandharvas* for *rākṣasas*, perhaps again stressing more dangerous aspects rather than the beauteous ones. Manuscript K4 adds that the *yakṣa*, *rākṣasa*, *gandharva*, and *nāga* maidens (*kanyā*) quickly hide (*paṇājire*) from Bhīma.

After this, Bhīma meets Hanumān, and they engage in battle with each other (3.146:49–3.150:28). While Bhīma nominally loses, he is praised by Hanumān for his valiance, and Hanumān agrees to help him on his quest. Initially he tries to deter Bhīma from joining in his quest. In 3.147:40 and its most relevant addition by manuscript group S, Hanumān states that the path Bhīma wants to take is divine and cherished by the *devas*, and Bhīma might get crushed or cursed by a *yakṣa* or *rākṣasa* if he treads it.[14] In a more off-topic discussion, Hanumān tells Bhīma that there were no *devas*, *dānavas*, *gandharvas*, *yakṣas*, *rākṣasas*, or *nāgas* (which manuscript T1 repaces with *kinnaras*) during the *kṛta yuga* or Golden Age (3.148:12).[15] When Bhīma is ultimately unpersuaded to quitting his quest, Hanumān tells Bhīma about the gardens of Kubera which are guarded by *yakṣas* and *rākṣasas*, and where one can only pick flowers after giving proper honour to the *devas* (3.149:22).[16]

After this, Bhīma goes on the move again and arrives at a lake near Mount Gandhamādana and the Saugandhika forest, which is the sporting region of Kubera. Again, this place is described as beautiful, often frequented by *gandharvas*, *apsarases*, *devas*, *ṛṣis*, *yakṣas*, *kimpuruṣas*, *rākṣasas*, and *kinnaras*, where the water tastes likes *amṛta* (3.151:7–8).[17] Later, when Bhīma wants to pick flowers and drink from the lake, he is stopped by Krodhavaśa *rākṣasas*, who state that *devarṣis*, *yakṣas*, *devas*, *gandharvas*, and *apsarases* have to ask permission from Kubera to drink from or play there (3.152:5).[18] A battle with the *rākṣasas* ensues in which the *yakṣas* are not mentioned, and which Bhīma wins (3.152–12-25). Later, it is stated that Bhīma defeated wide- eyed *yakṣas* by smashing their bodies, eyes, arms, thighs, and heads (3.153:24).[19] This does not fit in with the description of the actual battle, in which *yakṣas* are absent, which is perhaps why manuscript G1 and manuscript group Dc corrects *rākṣasān* for *yakṣān su-*.

When examining this second narrative, four markers are employed. The second marker is purely negative, since the *yakṣas* are guardians for Kubera's abode and enemies of Bhīma. This actually makes them supernatural entities who belong to the proper order, since they are underlings of the *deva* Kubera and also uphold proper rituals. Similarly, they

are entities associated with the beauty of Mount Gandhamādana, and referred to as means to indicate its beauty. However, both these last markers also read as negative in a couple of manuscripts where the *yakṣas* are more closely associated with dangerous supernatural entities, like *rākṣasas* and *piśācas*. In short, ambiguity is also found here because of the mixing of different markers (see Table 2).

**Table 2.** Evaluation of the *yakṣas* in the First War of the *Yakṣas*.

| Marker | Positive | Negative |
|---|---|---|
| Help/Hindrance | - | - |
| Protectors/Attackers | - | Guardians (3.147:40; 3.149:22); enemies (3.153:24) |
| Social order | *Yakṣas* among proper supernatural entities (3.146:23, 32–33); *yakṣas* uphold proper rituals (3.149:22; 3.152:5) | *Yakṣas* among improper supernatural entities (3.146:23, 32–33) |
| Beautiful/Gruesome | Beautiful mount Gandhamādana (3.146:23, 32–33; 3.151:7–8) | - |
| Central/Peripheral | Beautiful mount Gandhamādana (3.146:23, 32–33; 3.151:7–8) | Dangerous mount Gandhamādana (3.146:23, 32–33) |

### 3.3. The Yakṣas in the Second War of the Yakṣas

The Second War has seventeen textual places mentioning the *yakṣas*. The story begins with Janamejaya (a Kuru king to which the story of the Pāṇḍavas is told—one of the narrative frames building up the Mahābhārata) asking Vaiśaṃpāyana (the narrator of this story) to continue the narrative of Bhīma, whether he fought the *yakṣas* at the Himālaya mountains (where Mount Gandhamādana is located), and whether he met Kubera (3.157:3).[20] Vaiśaṃpāyana then continues the story. After being praised for his strength by Draupadī, Bhīma goes to chase the supernatural enemies from the mountain (3.157:18–24). When arriving at Mount Gandhamādana, its beauty is again described (3.157:35–40). After that, the *yakṣas*, *rākṣasas*, and *gandharvas*, with all of their hairs raised, start attacking Bhīma with various weapons. Bhīma cuts of their limbs, hands, and heads, after which the *yakṣas* utter sounds of fear and flee to the south, leaving their weapons behind (3.157:41–51).[21] The *rākṣasa* Maṇimāt insults the fleeing armies, and attacks Bhīma himself. Only in manuscript K4 do we have *yakṣarāt* instead of *rākṣasaḥ*, meaning that Maṇimāt is a king or commander of the *yakṣas* (3.157:52).[22] Attacking Bhīma with multiple weapons, he is eventually struck down, and falls down like a witch (*kṛti eva*), after which he flees (3.157:68).[23] In all other places, however, Maṇimāt is known as a *rākṣasa*, even within manuscript K4 (which changes his title in the earlier verse 52), showing again how fluid the border between *yakṣas* and *rākṣasas* can be.

After this, Bhīma returns to the other Pāṇḍavas. Yudhiṣṭhira condemns Bhīma's actions. It is contrary to the wishes of the king (Yudhiṣṭhira, that is), and hateful to the thirty *devas*, and therefore contrary to *dharma* (3.158:9–15). At the same time, *rākṣasas* report to Kubera that the foremost of *yakṣas* and *rākṣasas* have been slain by Bhīma. Everywhere, except in manuscripts D5, M1, Ś1, D1, D2, D3, and manuscript groups Dc and K, Kubera is named the overlord of *yakṣas* (*rājanyakṣādhipatim*) (3.158:16–19),[24] as well as in a later verse (21–22).[25]

This naturally angers Kubera, and he sets out after him. He is followed by many *yakṣas*,[26] who are described elaborately. They have reddened or bloodshot eyes (*raktākṣā*),[27] a golden hue (*hemasaṃkāśā*), huge bodies (*mahākāyā*), and accompanying strength (*mahābalāḥ*). Seven manuscripts replace this last point with *mahājavā* or great speed.[28] Manuscript group

S replaces this all with the statement that the *yakṣas* have a terrifying appearance (*ghoradarśānāḥ*) and follow Kubera.[29] The *yakṣas* are praised here for their abilities and are called great heroes (*javena mahatā vīrāḥ*) with swords[30] (3.158:25–29).[31] Next we learn that *yakṣas* are capable of flight and are nimble like birds (3.158:31).[32]

Then Kubera arrives before the Pāṇḍavas, who bow before him. This pleases Kubera immensely, and the *yakṣas* and *gandharvas* who accompanied him become pacified (3.158:32).[33] This is a surprising turn in the narrative, which will soon be explained. First, Kubera sits down on his seat Puṣpaka, and is surrounded by thousands of huge-bodied and pointy-eared *yakṣas* and *rākṣasas*, while there are also hundreds of *gandharvas* and *apsarases* present. The *yakṣas* and *rākṣasas* are also described as very swift (*mahājavā*), while four manuscripts (K1, K2, M1, and Ś1) replace this with very strong (*mahābalāḥ*) (3.158:35–37).[34] Kubera first addresses Yudhiṣṭhira. He allows the Pāṇḍavas to stay on Mount Gandhamādana. They should not regret the slaying of *yakṣas* and *rākṣasas*, for their deaths had been foreseen by the *devas* (*dṛṣṭaścāpi suraiḥ pūrva vināśo*; 3.158:43).[35] Then Kubera turns to Bhīma, and essentially repeats this message. He states that Bhīma only did what he did to please Draupadī, and additionally, with the battle he managed to freed Kubera from a curse (3.158:46).[36]

This curse is described as follows. The *devas* are to gather for a conclave at Kuśavatyā. Kubera goes there, surrounded by an extremely large number (*mahāpadmaśataistribhiḥ*) of *yakṣas* (and according to manuscript M1 also *rākṣasas*). These *yakṣas* have terrifying appearances and carry all kinds of weapons. Manuscript B4 omits their terrifying appearance (*ghorarūpāṇām*), exchanging it for an ability to change their shape at will (*kāmarūpāṇām*); this is the only place within the Araṇya Parva explicitly mentioning this ability (3.158:51).[37] While there, Maṇimāt (who was identified as a king of *yakṣas* in 3.157:52, but only according to manuscript K4) spits on *maharṣi* Agastya from the sky. Kubera is cursed for not stepping in: Maṇimāt and his army will be destroyed by a human, and Kubera will suffer from failing to prevent this. The curse, however, will be lifted once Kubera lays his eyes upon the slayer of his troops (3.158:52–59), which happened shortly before he sees Bhīma.

Kubera, being grateful, allows the Pāṇḍavas to live at the *āśrama* of *rājarṣi* Ārṣṭiṣeṇa on Mount Gandhamādana. The mountain is also inhabited by *gandharvas*, *yakṣas*, *rākṣasas*, and *alakās* (inhabitants of Kubera's residence Alakā; Monier-Williams 1899, p. 94), who will now protect the Pāṇḍavas. Yudhiṣṭhira has to keep Bhīma in check, however, because of his reputation for killing *yakṣas* and *rākṣasas* (3.159:11).[38] Later it is stated that Kubera's servants (*matpreṣyāḥ*) will provide the Pāṇḍavas with food (*anna*) and alcohol (*pāni*), but it is not completely certain whether these *matpreṣyāḥ* are *yakṣas* (3.159:14).[39] While this is an uncertain affiliation, later on it becomes clear that the *yakṣas* should accommodate all the Pāṇḍavas' desires (3.159:27).[40] After these statements, Kubera leaves, and his *yakṣas* and *rākṣasas* follow him on beautiful vehicles covered with checkered cushions and decorated with various jewels (3.159:29–31).[41] Now the Pāṇḍavas can safely spend the night on Mount Gandhamādana while being honoured by all *rākṣasas* (and *yakṣas* according to manuscripts K4, M1, and T1) (3.159:35).[42]

In this narrative, the role of the *yakṣas* is more complex than in the First War of the *Yakṣas*. The *yakṣas* inhabit all markers, and are only unambiguous as fulfillers of wishes (positive evaluation) for the first marker. For the second marker, we find shifting indications throughout the narrative. First, the *yakṣas* are enemies of the Pāṇḍavas, but later they are pacified and even become protectors. This is therefore not so much ambiguity as story progression. They are more concretely ambiguous in the other markers. For the third marker, we see more diversity. First of all, *yakṣas* belong to some kind of proper order by belonging to Kubera, while they also maintain the proper order by bringing honour to the Pāṇḍavas. In addition to that, *dharma* is disturbed by their slaughter. On the other side of the coin, however, the *yakṣas* can also be found grouped together with dangerous supernatural entities like *piśācas*, and may even insult *ṛṣis* by spitting on them.

In terms of appearance, they are mainly described as terrifying. They are sometimes associated with beauty, like their vehicles or when they enhance the beauty of Mount Gandhamādana. Additionally, they also have a golden hue (*hemasaṃkāśā*), but it is uncertain how this colour *hema* should be interpreted. For horses, it refers to a dark or brown colour (Monier-Williams 1899, p. 1304), but as a noun can refer to gold (p. 1305). This descriptor could be neutral instead of a descriptor like *suvarṇa*, another denomer for the golden colour (p. 1236). Next to the beauty of Mount Gandhamādana, it can also be a gravely dangerous place. With markers 2, 3, and 5 as definitely ambiguous, marker 1 being positively evaluated, and marker 4 being more undetermined, we can conclude that the *yakṣas*' appearance in the Second War of the *Yakṣas* is also ambiguous (see Table 3).

**Table 3.** Evaluation of the *yakṣas* in the Second War of the *Yakṣas*.

| Marker | Positive | Negative |
|---|---|---|
| Help/Hindrance | Perhaps provide food and drink (3.159:14); *yakṣas* gratify wishes (3.159:27) | - |
| Protectors/Attackers | *Yakṣas* pacified (3.158:32); *yakṣas* as protectors (3.159:11) | *Yakṣas* as enemies (3.157:3, 41–51, 52, 3.158:16–19) |
| Social order | Bhīma's slaying of *yakṣas* against *dharma* (3.158:9–15); *yakṣas* among Kubera (3.158:16–19, 21–22, 25–29, 51; 3.159:29–31); *yakṣas* honour Pāṇḍavas (3.159:35) | *Yakṣas* among improper supernatural entities (3.157:41–51); Maṇimāt spits on Agastya (3.158:51) |
| Beautiful/Gruesome | Golden hue? (3.158:25–29); beautiful vehicles (3.159:29–31) | Terrifying appearance (3.158:26 ms. S; 3.158:51) |
| Central/Peripheral | Beautiful Mount Gandhamādana (3.157:35–40) | Dangerous Mount Gandhamādana (3.157:3) |

### 3.4. The Yakṣas in the Story of the Drilling Wood

One of the more famous literary *yakṣas* is the one found in the story of the Drilling Wood. Most interestingly, this renowned *yakṣa* turns out to not be a *yakṣa* at all, as will be clarified below. In this story, the Pāṇḍavas are chasing a deer that stole fire drilling wood from a Brahmin who had been performing an *agnihotra*. The deer is quite elusive, and at the end the Pāṇḍavas lose track of it. They rest underneath a tree, which Nakula eventually climbs a tree in order to scout for water for his tired, thirsty brothers. He sees some trees near water and hears many cranes, a sure indicator of fresh, drinkable water. Yudhiṣṭhira asks Nakula to fetch some for them, so Nakula departs. Upon entering the lake's vicinity Nakula hears a voice from *antarikṣātsa*, the intermediate space between heaven and earth. This voice asks him to answer some questions before he drinks, since this is the voice's old territory. Nakula ignores this, drinks from the lake, and drops down as though dead. Then his twin brother Sahadeva goes there, likewise ignores the voice and collapses (3.295:7–296:19).

There is something rather paradoxical about this part of the *Mahābhārata* narrative, as noted by Shulman. This episode happens just before the Pāṇḍavas have to hide in exile at the court of Virāṭa, and at this point it becomes apparent that the normal order of things is lopsided. Earlier in the story, the Pāṇḍavas complain about their fate, and that such horrible things have happened to them despite being good people. This contradiction is doubled by the thirst-quenching water which apparently killed the Pāṇḍavas (Shulman 2001, p. 42).

Yudhiṣṭhira, becoming alarmed, sends over Arjuna next. He is struck by grief after seeing his brothers. He lifts his bow, but sees no creature, so he goes to drink. Again, a voice from *antarikṣātsa* states that he cannot take water, and needs to answer some questions

first. Arjuna states his intent to shoot the voice so it will not speak that way again, and shoots many arrows (3.296:20–29). The next line finally identifies the invisible voice:

> The *yakṣa* said: 'What does this shooting profit you, Pārtha [another name for Arjuna]? Answer my questions and drink. If you do not answer, you shall cease to be as soon as you drink!' (3.296:30)[43]

The *yakṣa*'s warning does not matter, however. Arjuna still drinks from the water while ignoring the *yakṣa*, and he too collapses (3.296:31).

With the situation becoming more and more dire, Yudhiṣṭhira sends out Bhīma. Upon arriving at the lake, Bhīma does not immediately panic like the others. He believes his fallen brothers to be an illusion by some *yakṣa* or *rākṣasa*, whom he needs to fight (3.296:35).[44] Bhīma is told by the *yakṣa* to not drink from the water, but Bhīma ignores this, drinks, and falls down as though dead. This *yakṣa* is described as *yakṣeṇamitatejasā* (or *yakṣeṇādbhutatejasā* in manuscript D2 and manuscript group Dc), meaning of great beauty, brightness, or vital power (Monier-Williams 1899, p. 454) (3.296:37–38).[45]

With Bhīma gone, it is Yudhiṣṭhira's turn. He heads towards the lake, and the trip is described as beautiful. All kinds of beautiful flora abound, and the lake is piled with gold (3.296:39–43). Once he arrives at the lake, he suspects foul play by Duryodhana, the eldest Kaurava brother and sworn enemy of the Pāṇḍavas. However, he discovers that the water does not seem to be poisoned. Yudhiṣṭhira, unlike the other brothers, decides to maintain proper form, and first performs ablutions in the pool (3.297:1–10). This pleases the *yakṣa*, who identifies itself as a crane (*baka*) living on the fish of the lake. The *baka-yakṣa* admits to having put a spell on the other Pāṇḍava brothers, who are therefore not dead but merely asleep. Yudhiṣṭhira must answer questions before he can collect any water (3.297:11–12).[46] That the *yakṣa* in this text identifies itself as a *baka* is quite significant. Such a bird is often regarded as a hypocrite, cheater, and rogue, known for its cunning and deceit. Next to that, there have been other unpleasant supernatural beings who have disguised themselves as a *baka*, like an *asura* and a *rākṣasa* (Monier-Williams 1899, p. 719). Finally, it is also a bird that is closely associated with death (Doniger-O'Flaherty 1976, p. 116).

Even in this ominous disguise, Yudhiṣṭhira is not deterred. Even more so, Yudhiṣṭhira is not convinced of the *yakṣa*'s self-identification as a crane (which he calls *śakuni*). Rather, he asks whether the *yakṣa* is the chief of the *rudras*, *vasus*, or *maruts* (3.297:13).[47] Yudhiṣṭhira states that his brothers are immensely strong and could not be felled by *devas*, *gandharvas*, *asuras*, *rākṣasas*, nor *yakṣas* (this last one only added by manuscript B3; 3.297:15).[48] Then there follows a very curious verse. Yudhiṣṭhira states that great curiosity, interest, even desire, has been aroused (*kautūhalaṃ mahajjātaṃ*), but at the same time terror or panic (*sādhvasaṃ*) has come over him—a perfect combination for *mysterium tremendum et fascinans*. Yudhiṣṭhira is trembling within his heart (*yena-asmi–udvij–hṛdyaḥ*), and has a headache, is with fever (*śirojvaraḥ*) or without (*śirorujaḥ* in manuscripts B3 and M1) (3.297:16–17).[49]

The *yakṣa* then identifies itself as a *yakṣa*, not a bird. He speaks with rough or uneven syllables (*paruṣākṣarām*) and with an ominous tone (*tāmaśivām*). After that, the *yakṣa* reveals its appearance to Yudhiṣṭhira. It has unusual or deformed eyes (*virūpākṣaṃ*), and a huge body as tall as a palmyra palm, unassailable like a mountain (*adhṛṣyam parvatopamam*), and blazing like the sun (*jvalanārkapratīkāśam*). The *yakṣa*'s voice roars deep like the clouds (*maghagambhīrayā vācā*) in a threatening manner (*tarjāyatam mahābalam*), which Van Buitenen translates as a thunderclap. Next to all this, we find the *yakṣa* to be standing in an interesting location. There are three different manuscript traditions denoting where the *yakṣa* takes refuge (*āśritya*). Manuscript D2 and manuscript group Dc have *sara*, which could mean liquid or cord. So, either the *yakṣa* is standing on a cord or in water. Manuscripts D4, D5, G3, and manuscript groups B and Dn have the *yakṣa* perching in a tree (*vṛkṣam*). Most manuscripts, however, locate the *yakṣa* on a *setu*, which is a dam or ridge that separates one plot of cultivated land from another. Here, again, just like with the *antarikṣātsa* or intermediate area between heaven and earth, we find the *yakṣa* associated with a liminal and therefore truly ambiguous position (3.297:18–21).[50]

The *yakṣa* claims that he killed the Pāṇḍava brothers because they drank from the pool when he explicitly forbade them from doing so. Yudhiṣṭhira, likewise, is only allowed to drink after answering questions (3.297:22–23).[51] Yudhiṣṭhira, surprisingly, responds to this stipulation by praising the *yakṣa* as a lord (*prabho*) or as a bull among male beings (*puruṣarṣabha* in manuscripts D2, D4, D6, G3, and manuscript groups B and Dn). Yudhiṣṭhira agrees to being questioned (3.297:24–25).[52] These questions are traditionally known as *praśnavyākaraṇa* or *brahmodya* (especially as they appear in Yajur Veda 23), but most commonly as *praśnottara-mālikā* or garland of questions (Misra 1981, p. 19). According to Agrawala, such questions are an integral part of *yakṣa* worship and mimic the type of questions asked by someone who is possessed by a *yakṣa* (*yakṣa–graha*) (Agrawala 1970, p. 195). While White sees the content of the questions as mere *yakṣa–abhidharma* or *yakṣa* scholasticism (White 2021, p. 143), Shulman treats them with more attention. He places this within the Upaniṣadic riddling tradition (as did Nīlakaṇṭha, one of the primary commentators on the Mahābhārata, did before him; Shulman 2001, p. 43), where the riddlee (the one answering questions) is under direct peril, since a wrong answer may lead to a swift death. In such a riddling game, there is a concealed answer which does not directly come to light. Such an answer can be deduced through the questions being asked, which relate to each other by means of cognitive mapping: the different categories and cosmological levels are meant to run parallel to each other (pp. 45–46). Shulman claims that the core of the questions are about ultimate reality (*brahman*), which is subsumed under the first answer (p. 44):

> The *yakṣa* said: 'What causes the sun to rise, and what are its companions? What makes it set, and on what is it founded? Yudhiṣṭhira said: *Brahman* makes the sun rise, and the *devas* are its companions. *Dharma* makes it set, and on truth it is founded' (3.297:26–27)[53]

In the story, Yudhiṣṭhira's answers must be taken as truthful and demonstrating his wisdom. In reality, however, it is quite likely that these lists of questions and answers were memorized (Shulman 2001, p. 45), as happens more often in riddling traditions (Kaivola-Bregenhøj 2001, p. 56).

After answering these questions (3.297:26–64), the *yakṣa* allows Yudhiṣṭhira to awaken one of his brothers. Yudhiṣṭhira picks Nakula, for he argues that his own mother (Kunti) still has a living son (Yudhiṣṭhira himself), while Madri is bereft of both Nakula and Sahadeva. The *yakṣa* is impressed with this choice, since he would have expected Yudhiṣṭhira to pick one of his full brothers. Because of this, he lifts the spell on the other Pāṇḍava brothers (3.297:65–74). Yudhiṣṭhira, however, is still not convinced of the *yakṣa*'s identity, and asks again who he is: a *deva*, one of the *vasus* or *rudras*, the chief of the *maruts*, Indra, their friend, or their father (3.298:2–5).[54] Now, finally, the *yakṣa* reveals its true identity: it is the *deva* Dharma, Yudhiṣṭhira's divine father. In the Mahābhārata, Yudhiṣṭhira is tested three times by Dharma in order to demonstrate *sanātāna dharma*. This was the first instance. The second test concerns a dog who cannot enter heaven, for Indra does not allow this. Yudhiṣṭhira therefore chooses to stay with the dog (17.1–3). The third test is found in 18.1–2. Here, the Kauravas celebrate in heaven, while the Pāṇḍavas suffer in hell. Yudhiṣṭhira decides to stay in hell with his kinsmen (Fitzgerald 2020, pp. 33–36). Because of Yudhiṣṭhira's valour, Dharma gives them more boons: the Brahmin will be able to worship Agni without interruption,[55] the Pāṇḍavas will be able to remain in exile without being recognized, and Yudhiṣṭhira will be freed from greed, folly, and anger, and his mind will always be inclined towards charity, austerity, and truth (3.298:11–25).

In this last narrative, we again see a lot of ambiguity. In terms of net balance, it seems that the *yakṣa* is more negatively evaluated here because of its initial threat to the Pāṇḍavas. With regard to the first marker, the *yakṣa* prevents the thirst of the Pāṇḍavas from being quenched on several occasions. Near the end of the narrative, however, the *yakṣa* restores the Pāṇḍavas to life and provides them with boons. While the ambiguity here is of a linear fashion (meaning that it gets resolved through the progression of the story), we are left

with mainly a negative portrayal of the second marker. Once resolved, the *yakṣa* turns out to be the *deva* Dharma, but as a *yakṣa* his focal behaviour lies in its attacks on the Pāṇḍavas by cursing the lake, playing tricks, and putting the Pāṇḍavas under a spell.

A similar but slightly more complex linear progression is also found with regard to the third marker. The *yakṣa* starts out as a *baka* or *śakuni*, which is an extremely treacherous bird who essentially curses four of the Pāṇḍava brothers. After this, it is associated with proper supernatural entities by Yudhiṣṭhira. After being identified as a *yakṣa*, the *yakṣa* asks questions about ultimate reality. Finally, after the trial, as the *deva* Dharma, there is the restoration of the ritual order: the Brahmin can perform his ritual again.

The fourth marker, then, shows true ambiguity. The *yakṣa* has *tejas*, which could be seen as a positive marker, but he also has a terrifyingand fascinating appearance as betrayed by Yudhiṣṭhira's reaction. Similarly, for the final marker, the lake is a beautiful place but is simultaneously highly dangerous. When we also take into account the other two locative moments of liminality (the *yakṣa* speaking from *antarikṣātsa* (multiple occurrences) and standing on the *setu* in 3.297:18–21), we can see some definite signs of ambiguity in this narrative (see Table 4).

**Table 4.** evaluation of the *yakṣa* in the story of the Drilling Woods.

| Marker | Positive | Negative |
|---|---|---|
| Help/Hindrance | *Yakṣa* cures all Pāṇḍavas (3.297:65–74); Dharma provides boons to Pāṇḍavas (3.298:11–25) | *Yakṣa* prevents thirst being quenched (3.296:30, 37–38; 3.297:12, 22–23) |
| Protecting/Attacking | - | *Yakṣa* cursed lake (3.296:30); *yakṣas* play tricks (3.296:35); *yakṣa* kills Pāṇḍavas (3.297:22–23) |
| Social order | *Yakṣa* associated with proper supernatural entities (3.297:13); *yakṣa* asking questions about ultimate reality (3.297:26–64); *yakṣa* as Dharma (3.298:6–25); restoration of ritual order (3.298: | *Yakṣa* as *baka* (3.296:11–12) |
| Beautiful/Gruesome | *Yakṣa* has *tejas* (3.296:37–38) | Terrifying but fascinating appearance (3.297:16–21) |
| Central/Peripheral | Beautiful lake (3.296:39–43) | Cursed lake (3.296:30) |

## 4. Conclusions

The goal of this article is twofold. First of all, scouring through the literature, an ideal type model has been devised with which one can examine ambiguity in supernatural entities. Five markers have been found for positive and negative evaluations of supernatural species. The first marker considers whether the supernatural being aids humans in fulfilling desires and needs, or prevents such fulfilment. The second marker examines whether the supernatural beings protect or attack humans. The third marker determines whether the supernatural beings fall under the same order as humans, or if they seek to destroy that order. The fourth marker zooms in on the appearance of the supernatural entity, and whether they conform to cultural ideas of beauty and decency, or if they break with these. Finally, the last marker examines the location (either close by or far away) of the supernatural being.

This model has been used to analyse four narratives within the Araṇya Parva of the Mahābhārata: the story of Nala and Damayantī (3.50–3.78), the First War of the *Yakṣas* (3.146–3.153), the Second War of the *Yakṣas* (3.157–159), and the story of the Drilling Woods

(3.295–3.299). In all four of these narratives, the *yakṣas* are found to be ambiguous; that is, the *yakṣas* in these narratives have a combination of positive and negative markers. All of the markers have been employed to determine the *yakṣas'* ambiguity. One can therefore conclude that *yakṣas* are utilized in their ambiguity in the Araṇya Parva of the Mahābhārata. In addition to that, we can now more precisely determine what this ambiguity looks like.

At the same time, we should note the difference between true ambiguity (meaning that both positive and negative markers are present and active at the same narrative moment) as well as ambiguity caused by narrative development. We have seen a couple of examples this last group. First of all, in the story of Nala and Damayantī, there is a shift from the positively evaluated *yakṣiṇī*-Damayantī who can potentially fulfill wishes and protect the merchant caravan, to one who destroys and causes suffering. Secondly, in the Second War of the *Yakṣas*, we start with *yakṣas* as antagonists who eventually become pacified and even protectors of the Pāṇḍavas. Finally, in the story of the Drillling Woods, there is a shift from the *yakṣa* as an entity that prevents the fulfilment of desires (quenching thirst) to one who grants boons (reviving the Pāṇḍavas); one can also see a gradual pacifying shift from *baka* to *yakṣa* and finally to the *deva* Dharma. While there is now a model which serves to determine and analyse ambiguity, we should correspondingly not lose sight of the ambiguous nature of supernatural entities and their tendency to confound any clear-cut analysis of them.

**Funding:** This research received no external funding.

**Data Availability Statement:** The data presented in this study are available in this article, and is specified in the footnotes.

**Conflicts of Interest:** The author declares no conflict of interest.

## Notes

[1] Translation based on Van Buitenen (1975, p. 341); Sanskrit: Sukthankar (1942, p. 204):

kāsi kasyāsi kalyāṇi | kiṃ vā mṛgayase vane
tvāṃ dṛṣṭvā vyathitāḥ smeha | kaścattvamasi mānuṣī |113
vada satyaṃ vanasyāsya | parvatasyātha vā diśaḥ
devatā tvaṃ hi kalyāṇi | tvāṃ vayaṃ śaraṇaṃ gatāḥ |114
yakṣī vā rākṣasi vā tvam | utāho'si varāṅganā
sarvathā kuru naḥ svasti | rakṣasvāsmānanindite |115
yathāyaṃ sarvathā sārthaḥ | kṣemī śīdhramito vrajet
tathā vidhatsva kalyāṇi | tvāṃ vayaṃ śaraṇaṃ gatāḥ |116

[2] Translation based on Ganguli (1884, p. 141); Sanskrit: Sukthankar (1942, p. 1057):

yāsāvadya mahāsārthe | nārīvonmattadarśanā |58
praviṣṭā vikṛtākārā | kṛtvā rūpamamānuṣam |59
tayeyaṃ vihitā pūrvaṃ | māyā paramadārūṇā |60
rākṣasī vā piśācī vā | yakṣī vātibhayaṃkarī |61
tasyāḥ sarvamidaṃ pāpaṃ | nātra kāryā vicāraṇā |62
yadi paśyām tāṃ pāpāṃ | sārthadhgīṃ naukaduḥkhadām |63
loṣṭakaiḥ pāśubhiścaiva | tṛṇaiḥ kāṣṭhaiśca muṣṭibhiḥ |64
avaśyameva hantavyā | sā sārthasya tu kṛcchradā |65

[3] This phrase is not coined by Otto himself, but it is commonly used as a shorthand paraphrase for Otto's main idea; see Otto (1917).

[4] As has been noted by many scholars, among them (Ballard 1981, p. 26; Bhattacharya 2022, pp. 9, 12, 17; Erndl 1989, pp. 239–40; Hansen 2001, pp. 22, 24; Hiltebeitel 1989b, p. 1; Hiltebeitel 1989a, p. 357; Kieckhefer 1998, pp. 154–55; Leach 1982, p. 215; Page 2011, p. 134; Sanchez 2021, p. 209; Shulman 1989, pp. 43, 59–60; Sparing 1984, p. 129; White 2003, p. 47; and White 2021, p. 1).

[5] Doniger-O'Flaherty 1976 is one of the main sources in this section. However, her book is quite unclear in delineating who the 'gods' and 'demons' are. It becomes apparent that the 'gods' are *devas*, but it is by no means always clear whether the 'demons' are solely the *asuras*, or could also include beings like *yakṣas*, *rākṣasas*, *nāgas* and others.

[6] Manuscript K3 has *rājendra*, which changes 'people' into 'emperor'.

[7] Translation based on Van Buitenen (1975, p. 323); Sanskrit: Sukthankar (1942, p. 166):

na deveṣu na yakṣeṣu | tādṛṛg rūpavatī kva cit
mānuṣeṣv api cānyeṣu | dṛṛṣṭapūrvā na ca śrutā |13

8　　Translation based on Van Buitenen (1975, p. 326); Sanskrit: Sukthankar (1942, p. 172):
　　　aho rūpam aho kāntir ǀ aho dhairyaṁ mahātmanaḥ
　　　ko 'yaṁ devo nu yakṣo nu ǀ gandharvo nu bhaviṣyati ǀ16

9　　Translation: Van Buitenen (1975, p. 341); Sanskrit: Sukthankar (1942, p. 205):
　　　kuñjaradvīpimahiṣa ǀ śārdūlarkṣamṛrgān api
　　　paśyāmy asmin vane kaṣṭe ǀ amanuṣyaniṣevite
　　　tathā no yakṣād adya ǀ maṇibhadraḥ prasīdatu ǀ123

10　　Translation: Ganguli (1884, p. 141); Sanskrit: Sukthankar (1942, p. 1055 (appendix 10)):
　　　nūnaṁ na pūjito 'smābhir ǀ maṇibhadro mahāyaśāḥ ǀ60
　　　tathā yakṣādhipaḥ śrīmān ǀ na ca vaiśravaṇaḥ prabhuḥ ǀ61

11　　For the Araṇya Parva of the Mahābhārata these are 3.41:14; lines 60 and 61 of appendix 10 of the critical edition, substitution for 3.62:1–17; 3.81:42; 3.140:4–8; 3.151:7–8; 3.152:5; 3.156:26; 3.157:52–70; 3.158:16–19; 3.158:21–22; 3.158:29; 3.258:16; 3.265:23; and 3.275:18.

12　　Translation: Van Buitenen (1975, p. 499); Sanskrit: Sukthankar (1942, p. 473):
　　　sa yakṣagandharvasura ǀ brahmarṣigaṇasevitam
　　　viloḍayām āsa tadā ǀ puṣpahetor ariṁdamaḥ ǀ23

13　　Translation: Van Buitenen (1975, p. 500); Sanskrit: Sukthankar (1942, p. 474):
　　　priyapārśvopaviṣṭābhir ǀ vyāvṛttābhir viceṣṭitaiḥ
　　　yakṣagandharvayoṣābhir ǀ adṛśyābhir nirīkṣitaḥ ǀ32
　　　navāvatāraṁ rūpasya ǀ vikrīṇann iva pāṇḍavaḥ
　　　cacāra ramaṇīyeṣu ǀ gandhamādanasānuṣu ǀ33

14　　Translation based on Van Buitenen (1975, p. 504); Sanskrit: Sukthankar (1942, p. 485):
　　　ayaṁ ca mārgo martyānām ǀ agamyaḥ kurunandana
　　　tato 'haṁ ruddhavān mārgaṁ ǀ tavemaṁ devasevitam
　　　tvām anena pathā yāntaṁ ǀ yakṣo vā rākṣaso 'pi vā
　　　dharṣayed vā śaped vāpi ǀ mā kaś cid iti bhārata ǀ40 according to S

15　　Translation: Van Buitenen (1975, p. 504); Sanskrit: Sukthankar (1942, p. 486):
　　　devadānavagandharva ǀ yakṣarākṣasapannagāḥ
　　　nāsan kṛtayuge tāta ǀ tadā na krayavikrayāḥ ǀ12

16　　Translation: Van Buitenen (1975, p. 507); Sanskrit: Sukthankar (1942, p. 491):
　　　eṣa panthāḥ kuruśreṣṭha ǀ saugandhikavanāya te
　　　drakṣyase dhanadodyānaṁ ǀ rakṣitaṁ yakṣarākṣasaiḥ ǀ22

17　　Translation: Van Buitenen (1975, p. 510); Sanskrit: Sukthankar (1942, p. 497):
　　　ākrīḍaṁ yakṣarājasya ǀ kuberasya mahātmanaḥ
　　　gandharvair apsarobhiś ca ǀ devaiś ca paramārcitām ǀ7
　　　sevitām ṛṣibhir divyāṁ ǀ yakṣaiḥ kiṁpuruṣais tathā
　　　rākṣasaiḥ kiṁnaraiś caiva ǀ guptāṁ vaiśravaṇena ca ǀ8

18　　Translation: Van Buitenen (1975, p. 511); Sanskrit: Sukthankar (1942, p. 498):
　　　devarṣayas tathā yakṣā ǀ devāś cātra vṛkodara
　　　āmantrya yakṣapravaraṁ ǀ pibanti viharanti ca
　　　gandharvāpsarasaś caiva ǀ viharanty atra pāṇḍava ǀ5

19　　Translation: Van Buitenen (1975, p. 513); Sanskrit: Sukthankar (1942, p. 503):
　　　taṁ ca bhīmaṁ mahātmānaṁ ǀ tasyās tīre vyavasthitam
　　　dadṛśur nihatāṁś caiva ǀ yakṣān suvipulekṣaṇān ǀ24

20　　Translation: Van Buitenen (1975, p. 525); Sanskrit: Sukthankar (1942, p. 528):
　　　vistareṇa ca me śaṁsa ǀ bhīmasenaparākramam
　　　yad yac cakre mahābāhus ǀ tasmin haimavate girau
　　　na khalv āsīt punar yuddhaṁ ǀ tasya yakṣair dvijottama ǀ3

21　　Translation: Van Buitenen (1975, p. 527); Sanskrit: Sukthankar (1942, p. 531–32):
　　　tataḥ saṁhṛṣṭaromāṇaḥ ǀ śabdaṁ tam abhidudruvuḥ
　　　yakṣarākṣasagandharvāḥ ǀ pāṇḍavasya samīpataḥ ǀ41
　　　gadāparighanistriṁśa ǀ śaktiśūlaparaśvadhāḥ
　　　pragṛhītā vyarocanta ǀ yakṣarākṣasabāhubhiḥ ǀ42
　　　tataḥ pravavṛrte yuddhaṁ ǀ teṣāṁ tasya ca bhārata
　　　taiḥ prayuktān mahākāyaiḥ ǀ śaktiśūlaparaśvadhān
　　　bhallair bhīmaḥ praciccheda ǀ bhīmavegatarais tataḥ ǀ43
　　　antarikṣacarāṇāṁ ca ǀ bhūmiṣṭhānāṁ ca garjatām

śarair vivyādha gātrāṇi | rākṣasānāṁ mahābalaḥ |44
sā lohitamahāvṛṣṭir | abhyavarṣan mahābalam
kāyebhyaḥ pracyutā dhārā | rākṣasānāṁ samantataḥ |45
bhīmabāhubalotsṛṣṭair | bahudhā yakṣarakṣasām
vinikṛttāny adṛśyanta | śarīrāṇi śirāṁsi ca |46
pracchādyamānaṁ rakṣobhiḥ | pāṇḍavaṁ priyadarśanam
dadṛśuḥ sarvabhūtāni | sūryam abhragaṇair iva |47
sa raśmibhir ivādityaḥ | śarair arinighātibhiḥ
sarvān ārchan mahābāhur | balavān satyavikramaḥ |48
abhitarjayamānāś ca | ruvantaś ca mahāravān
na mohaṁ bhīmasenasya | dadṛśuḥ sarvarākṣasāḥ |49
te śaraiḥ kṣatasarvāṅgā | bhīmasenabhayārditāḥ
bhīmam ārtasvaraṁ cakrur | viprakīrṇamahāyudhāḥ |50
utsṛjya te gadāśūlān | asiśaktiparaśvadhān
dakṣiṇāṁ diśam ājagmus | trāsitā dṛḍhadhanvanā |51

[22] Translation: Van Buitenen (1975, p. 527); Sanskrit: Sukthankar (1942, p. 532):
tatra śūlagadāpāṇir | vyūḍhorasko mahābhujaḥ
sakhā vaiśravaṇasyāsīn | maṇimān nāma rākṣasaḥ |52

[23] Translation: Van Buitenen (1975, p. 528); Sanskrit: Sukthankar (1942, p. 533):
sendrāśanir ivendreṇa | visṛṣṭā vātaraṁhasā
hatvā rakṣaḥ kṣitiṁ prāpya | kṛtyeva nipapāta ha |68

[24] Translation: Van Buitenen (1975, pp. 528–29); Sanskrit: Sukthankar (1942, p. 535):
nyastaśastrāyudhāḥ śrāntāḥ | śoṇitāktaparicchadāḥ
prakīrṇamūrdhajā rājan | yakṣādhipatim abruvan |16
gadāparighanistriṁśa | tomaraprāsayodhinaḥ
rākṣasā nihatāḥ sarve | tava deva puraḥsarāḥ |17
pramṛdya tarasā śailaṁ | mānuṣeṇa dhaneśvara
ekena sahitāḥ saṁkhye | hatāḥ krodhavaśā gaṇāḥ |18
pravarā rakṣasendrāṇāṁ | yakṣāṇāṁ ca dhanādhipa
śerate nihatā deva | gatasattvāḥ parāsavaḥ |19

[25] Translation: Van Buitenen (1975, p. 529); Sanskrit: Sukthankar (1942, p. 535):
sa tac chrutvā tu saṁkruddhaḥ | sarvayakṣagaṇādhipaḥ
kopasaṁraktanayanaḥ | katham ity abravīd vacaḥ |21
dvitīyam aparādhyantaṁ | bhīmaṁ śrutvā dhaneśvaraḥ
cukrodha yakṣādhipatir | yujyatām iti cābravīt |22

[26] Different manuscript traditions provide different formulations for the huge numbers in 3.158:28. Manuscripts D3, D5, K1, K3, and K4 have *śatāvarāḥ*; K2 has *satāsataḥ*; D1, D2, D4, D6, and manuscript groups B, Dc, and Dn have *daśaśatāvarāḥ*, which is closer to the term found in all remaining manuscripts (*daśaśatāyutāḥ*).

[27] Manuscript T1 replaces this with *rākṣasā*, which is the only manuscript to add the *rākṣasas* in this passage.

[28] D1, D2, D3, K1, K2, K3, and Ś1.

[29] Sanskrit: Sukthankar (1942, p. 536):
anujagmurmahātmānaṁ | dhanadaṁ ghoradarśanāḥ

[30] *baddhanistriṁśā* (and in K1 and K2 *ghṛtanistriṁśā*), the last element needs to be corrected to *niḥtrimśā*.

[31] Translation based on Van Buitenen (1975, p. 529); Sanskrit: Sukthankar (1942, p. 536):
śobhamānā rathe yuktās | tariṣyanta ivāśugāḥ
harṣayām āsur anyonyam | iṅgitair vijayāvahaiḥ |25
sa tam āsthāya bhagavān | rājarājo mahāratham
prayayau devagandharvaiḥ | stūyamāno mahādyutiḥ |26
taṁ prayāntaṁ mahātmānaṁ | sarvayakṣadhanādhipam
raktākṣā hemasaṁkāśā mahākāyā mahābalāḥ |27
sāyudhā baddhanistriṁśā | yakṣā daśaśatāyutāḥ
javena mahatā vīrāḥ | parivāryopatasthire |28
taṁ mahāntam upāyāntaṁ | dhaneśvaram upān
tikedadṛśur hṛṣṭaromāṇaḥ | pāṇḍavāḥ priyadarśanam |29

[32] Translation: Van Buitenen (1975, p. 529); Sanskrit: Sukthankar (1942, p. 536):
te pakṣiṇa ivotpatya | gireḥ śṛṅgaṁ mahājavāḥ
tasthus teṣāṁ samabhyāśe | dhaneśvarapuraḥsarāḥ |31

33    Translation: Van Buitenen (1975, p. 529); Sanskrit: Sukthankar (1942, p. 536):

tatas taṁ hṛṣṭamanasaṁ | pāṇḍavān prati bhārata
samīkṣya yakṣagandharvā | nirvikārā vyavasthitāḥ |32

34    Translation based on Van Buitenen (1975, p. 529); Sanskrit: Sukthankar (1942, pp. 536–37):

śayyāsanavaraṁ śrīmat | puṣpakaṁ viśvakarmaṇā
vihitaṁ citraparyantam | ātiṣṭhata dhanādhipaḥ |35
tam āsīnaṁ mahākāyāḥ | śaṅkukarṇā mahājavāḥ
upopaviviśur yakṣā | rākṣasāś ca sahasraśaḥ |36
śataśaś cāpi gandharvās | tathaivāpsarasāṁ gaṇāḥ
parivāryopatiṣṭhanta | yathā devāḥ śatakratum |37

35    Translation: Van Buitenen (1975, p. 530); Sanskrit: Sukthankar (1942, p. 537):

vrīḍā cātra na kartavyā | sāhasaṁ yad idaṁ kṛtam
dṛṣṭaś cāpi suraiḥ pūrvaṁ | vināśo yakṣarakṣasām |43

36    Translation: Van Buitenen (1975, p. 530); Sanskrit: Sukthankar (1942, p. 537):

mām anādṛtya devāṁś ca | vināśaṁ yakṣarakṣasām
svabāhubalam āśritya | tenāhaṁ prītimāṁs tvayi |46

37    Translation based on Van Buitenen (1975, p. 530); Sanskrit: Sukthankar (1942, p. 538):

devatānām abhūn mantraḥ | kuśavatyaṁ nareśvara
vṛtas tatrāham agamaṁ | mahāpadmaśatais tribhiḥ
yakṣāṇāṁ ghorarūpāṇāṁ | vividhāyudhadhāriṇām |51

38    Translation based on Van Buitenen (1975, p. 531); Sanskrit: Sukthankar (1942, p. 540):

alakāḥ saha gandharvair | yakṣaiś ca saha rākṣasaiḥ
manniyuktā manuṣyendra | sarve ca girivāsinaḥ
rakṣantu tvā mahābāho | sahitaṁ dvijasattamaiḥ |11

39    Translation: Van Buitenen (1975, p. 531); Sanskrit: Sukthankar (1942, p. 540):

tathaiva cānnapānāni | svādūni ca bahūni ca
upasthāsyanti vo gṛhya | matpreṣyāḥ puruṣarṣabha |14

40    Translation based on Van Buitenen (1975, p. 532); Sanskrit: Sukthankar (1942, p. 540):

sveṣu veśmasu ramyeṣu | vasatāmitratāpanāḥ
kāmān upahariṣyanti | yakṣā vo bharatarṣabhāḥ |27

41    Translation based on Van Buitenen (1975, p. 532); Sanskrit: Sukthankar (1942, p. 540):

evam uttamakarmāṇam | anuśiṣya yudhiṣṭhiram
astaṁ girivaraśreṣṭhaṁ | prayayau guhyakādhipaḥ |29
taṁ paristomasaṁkīrṇair | nānāratnavibhūṣitaiḥ
yānair anuyayur yakṣā | rākṣasāś ca sahasraśaḥ |30
pakṣiṇām iva nirghoṣaḥ | kuberasadanaṁ prati
babhūva paramāśvānām | airāvatapathe yatām |31

42    Translation based on Van Buitenen (1975, p. 532); Sanskrit: Sukthankar (1942, p. 542):

pāṇḍavāpi mahātmānas | teṣu veśmasu tāṁ kṣapām
sukham ūṣur gatodvegāḥ | pūjitā yakṣarākṣasaiḥ |35 according to manuscripts K4, M1, and T1

43    Translation based on Van Buitenen (1975, p. 798); Sanskrit: Sukthankar (1942, p. 1023):

yakṣa uvāca |
kiṁ vighātena te pārtha | praśnān uktvā tataḥ piba
anuktvā tu tataḥ praśnān | pītvaiva na bhaviṣyasi |30

44    Translation based on Van Buitenen (1975, p. 798); Sanskrit: Sukthankar (1942, p. 1024):

tān dṛṣṭvā duḥkhito bhīmas | tṛṣayā ca prapīḍitaḥ
amanyata mahābāhuḥ | karma tad yakṣarakṣasām
sa cintayām āsa tadā | yoddhavyaṁ dhruvam adya me |35

45    Translation based on Van Buitenen (1975, p. 798); Sanskrit: Sukthankar (1942, p. 1024):

yakṣa uvāca |
mā tāta sāhasaṁ kārṣīr | mama pūrvaparigrahaḥ
praśnān uktvā tu kaunteya | tataḥ piba harasva ca |37
vaiśaṁpāyana uvāca
evam uktas tato bhīmo | yakṣeṇāmitatejasā
avijñāyaiva tān praśnān | pītvaiva nipapāta ha |38

46    Translation based on Van Buitenen (1975, p. 799); Sanskrit: Sukthankar (1942, p. 1026):

yakṣa uvāca |

ahaṁ bakaḥ śaivalamatsyabhakṣo ǀ mayā nītāḥ pretavaśaṁ tavānujāḥ
tvaṁ pañcamo bhavitā rājaputra ǀ na cet praśnān pṛcchato vyākaroṣi ǀ11
mā tāta sāhasaṁ kārṣīr ǀ mama pūrvaparigrahaḥ
praśnān uktvā tu kaunteya ǀ tataḥ piba harasva ca ǀ12

47 Translation based on Van Buitenen (1975, p. 799); Sanskrit: Sukthankar (1942, p. 1027):
yudhiṣṭhira uvāca ǀ
rudrāṇaṁ vā vasūnāṁ vā ǀ marutāṁ vā pradhānabhāk
pṛcchāmi ko bhavān devo ǀ naitac chakuninā kṛtam ǀ13

48 Translation based on Van Buitenen (1975, pp. 799–800); Sanskrit: Sukthankar (1942, p. 1027):
atīva te mahat karma ǀ kṛtaṁ ca balanāṁ vara
yan na devā na gandharvā ǀ nāsurā yakṣarākṣasāḥ
viṣaheran mahāyuddhe ǀ kṛtaṁ te tan mahādbhutam ǀ15 according to manuscript B3.

49 Translation based on Van Buitenen (1975, p. 800); Sanskrit: Sukthankar (1942, p. 1027):
na te jānāmi yat kāryaṁ ǀ nābhijānāmi kāṅkṣitam
kautūhalaṁ mahaj jātaṁ ǀ sādhvasaṁ cāgataṁ mama ǀ16
yenāsmy udvignahṛdayaḥ ǀ samutpannaśirojvaraḥ
pṛcchāmi bhagavaṁs tasmāt ǀ ko bhavān iha tiṣṭhati ǀ17

50 Translation based on Van Buitenen (1975, p. 800); Sanskrit: Sukthankar (1942, p. 1027):
yakṣa uvāca ǀ
yakṣo 'ham asmi bhadraṁ te ǀ nāsmi pakṣī jalecaraḥ
mayaite nihatāḥ sarve ǀ bhrātaras te mahaujasaḥ ǀ18
vaiśaṁpāyana uvāca ǀ
tatas tām aśivāṁ śrutvā ǀ vācaṁ sa paruṣākṣarām
yakṣasya bruvato rājann ǀ upakramya tadā sthitaḥ ǀ19
virūpākṣaṁ mahākāyaṁ ǀ yakṣaṁ tālasamucchrayam
jvalanārkapratīkāśam ǀ adhṛṣyaṁ parvatopamam ǀ20
setum āśritya tiṣṭhantaṁ ǀ dadarśa bharatarṣabhaḥ
meghagambhīrayā vācā ǀ tarjayantaṁ mahābalam ǀ21

51 Translation based on Van Buitenen (1975, p. 800); Sanskrit: Sukthankar (1942, p. 1027–28):
yakṣa uvāca ǀ
ime te bhrātaro rājan ǀ vāryamāṇā mayāsakṛt
balāt toyaṁ jihīrṣantas ǀ tato vai sūditā mayā ǀ22
na peyam udakaṁ rājan ǀ prāṇān iha parīpsatā
pārtha mā sāhasaṁ kārṣīr ǀ mama pūrvaparigrahaḥ
praśnān uktvā tu kaunteya ǀ tataḥ piba harasva ca ǀ23

52 Translation based on Van Buitenen (1975, p. 800); Sanskrit: Sukthankar (1942, p. 1028):
yudhiṣṭhira uvāca ǀ
naivāhaṁ kāmaye yakṣa ǀ tava pūrvaparigraham
kāmaṁ naitat praśaṁsanti ǀ santo hi puruṣāḥ sadā ǀ24
yadātmanā svam ātmānaṁ ǀ praśaṁset puruṣaḥ prabho
yathāprajñaṁ tu te praśnān ǀ prativakṣyāmi pṛccha mām ǀ25

53 Translation: Van Buitenen (1975, p. 800); Sanskrit: Sukthankar (1942, p. 1028):
yakṣa uvāca ǀ
kiṁ svid ādityam unnayati ǀ ke ca tasyābhitaś carāḥ
kaś cainam astaṁ nayati ǀ kasmiṁś ca pratitiṣṭhati ǀ26
yudhiṣṭhira uvāca
brahmādityam unnayati ǀ devās tasyābhitaś carāḥ
dharmaś cāstaṁ nayati ca ǀ satye ca pratitiṣṭhati ǀ27

54 Translation: Van Buitenen (1975, p. 804); Sanskrit: Sukthankar (1942, p. 1034):
yudhiṣṭhira uvāca ǀ
sarasy ekena pādena ǀ tiṣṭhantam aparājitam
pṛcchāmi ko bhavān devo ǀ na me yakṣo mato bhavan ǀ2
vasūnāṁ vā bhavān eko ǀ rudrāṇām atha vā bhavan
atha vā marutāṁ śreṣṭho ǀ vajrī vā tridaśeśvaraḥ ǀ3
mama hi bhrātara ime ǀ sahasraśatayodhinaḥ
na taṁ yogaṁ prapaśyāmi ǀ yena syur vinipātitāḥ ǀ4
sukhaṁ prativibuddhānām ǀ indriyāṇy upalakṣaye
sa bhavān suhṛd asmākam ǀ atha vā naḥ pitā bhavan ǀ5

[55] It was actually Dharma disguised as a deer who stole the fire drilling sticks, showcasing that even the *devas* can be ambiguous figures.

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
