# Peer review of "Ka asi kasya asi, kalyāṇi?’ The Ambiguity of the yakṣas in the Araṇya Parva of the Mahābhārata"

_religions, doi:10.3390/rel14010037_

Round 1

Reviewer 1 Report

For the title, I was always taught to break apart the sandhi when quoting a Sanskrit verse: "Ka asi? Kasya asi?" instead of "Kāsi Kasyāsi."

The abstract needs editing. "The model for determining supernatural ambiguity utilizes five markers, which can either be positively or negatively evaluated" makes no sense, since each of the five markers is a pair of opposites. It should read, "which appear in either a positive or negative aspect." Also, the verbs in the pairs should match, e.g., on p. 1, line 10, "belonging" goes with "rejecting," not "reject."

On p. 2, line 37-8, what does “in association refer to?” What association? With what or whom? With each other? If so, let’s be clear about it.

On p. 2, line 52, I would not call a piśacī a ghost, but a ghoul.

On p. 4-5, lines 125-141, what does “constructed from” mean? The author need to explain this fully as it is the core of the paper.

On p. 8 (and a few other places) “next to that” is used when the author should use “in addition to that.”

On p. 10, lines 422-424, the author takes Holtzmann’s view of the epic as authoritative without discussing the many arguments against it.

At least in figure 1 (and preferably in all of the figures) the markers need to be restated rather than just referred to with the numbers 1-5.

On p. 13, line 549, “war on the” should be “war of the”

On p. 13, line 551, the author should use a term besides “well-reaking.”

There is a simpler way to state the author’s 5 areas of ambiguity: 1) help/hindrance; 2) protectors/attackers; 3) human-like social order/distinctive social order; 4) beautiful/gruesome; 5) central/peripheral.

Author Response

Dear reviewer,

Thank you for your comments on my article. I have essentially incorporated all of your notes. I am especially thankful for your suggestion on restating my five areas of ambiguity in a simpler way - I really appreciate that as a non-native English speaker. Additionally, a native speaker has reviewed the whole text and provided many suggestions for better phrasing.

Next to the adaptations based on your comments, I have also added a more elaborate section on Max Weber's ideal type, as suggested by another reviewer.

Thank you for your work, and all the best!

Reviewer 2 Report

The paper promises an innovative examination of a well-studied subject, the yaksas. 

Uses analytical frameworks and parameters that were normative to an earlier era, but criticized recently. But the information on hand is too small to provide adequate information to provide or prove/disprove a theory. It would be more useful to focus on the information on hand (two stories of the Aranyaparva) rather than the constructed frameworks of archetypes used in this paper.

1. The title does not justify the paper. The title is drawn from a confused statement with reference to Damayanti, who is not a yaksa/yaksini. The paper's central concern is Yaksas and their relationship with humans and others. However, contribution to this aspect is limited as more time and space is spent on unrelated information. Damayanti is a human. She was assumed as a yaksi, goddess, etc., in confusion, but her identity is always clearly stated in the Mahabharata. Therefore, beginning the paper with her story and taking the title from the confused statement uttered to her is misleading. The story of Damayanti should be eliminated from this paper since it brings confusion rather than clarity to the subject. 

2. using Max Weber's ideal type and then proposing markers to understand the characteristics of Yaksas as a way to decide ambiguity or clarity needs extensive informational resources. Analysis of two short stories is too little data to attain this goal.

3. Weber's ideal type is invoked but no references to the original theory, its application, or criticisms in relation to religions are provided. 

4. incorporates assumptions regarding chronology and classification of Hinduism- caution should be employed with a number of terms such as 'Brahminical village Hinduism'. 

Author Response

Dear reviewer,

Thank you for your comments. As per your suggestion, I have added an extensive section on Weber's ideal type and the manner in which I have utilized it in this research. I understand your remarks on perhaps foregoing my ideal type construction, but I am not inclined to do so. I have written about the heuristic value of the ideal type model, which I believe should be more convincing. Next to that, I have other reasons to hold onto this heuristic model, but I am unsure whether I can reveal those without also revealing my identity.

As to your comment about Damayanti, here I also disagree. I believe it is an apt inclusion, since it demonstrates one of the ways in which the image of the yaksha is used. Indeed, Damayanti is very clearly not a yakshini, but I believe it is quite telling that she is mistaken for one. I have added arguments in the introductory section of the article to support the inclusion of the narrative.

As to your fourth point, you are absolutely correct. I have added more nuances and removed most references to any kind of temporal division. 

Next to all this, a native speaker has revised my English prose, which has increased the clarity of some of my arguments drastically.

Thank you for your efforts, and all the best!

Round 2

Reviewer 2 Report

The paper is exceptionally well-written. 

Even though the inclusion of the story of Damayanti seems out of place, the author provided explanations about her being human, which is helpful in alleviating confusion. However, referring to Damayanti as yakṣi Damayanti or yakṣini Damayanti is not factual, which the paper does throughout this paper, including the conclusion. The text of the Mahabharata does not identify her explicitly as yakṣi, but only uses the terms, yakṣi, rakṣisi, or pisachi, as an indication of confusion of the merchants. This story is suitable to be discussed as an adjunct to the central argument of the paper. However, if the author does not like to do so, it is important not to refer to her as yakṣi Damayanti. If the author must keep this story and analysis, it is important that it is done with caution. Identifying one class of beings as another class of beings might mar the valuable research insights this paper brings forward.